# The HIV-1 latent reservoir is largely sensitive to circulating T cells

Joanna A Warren[1], Shuntai Zhou[2,3], Yinyan Xu[1], Matthew J Moeser[2,3], Daniel R MacMillan[4], Olivia Council[1,2], Jennifer Kirchherr[5], Julia M Sung[5,6], Nadia R Roan[7,8], Adaora A Adimora[5], Sarah Joseph[1,6], JoAnn D Kuruc[5,6], Cynthia L Gay[5,6], David M Margolis[1,3,5,6], Nancie Archin[5,6], Zabrina L Brumme[4,9], Ronald Swanstrom[1,2,3,6], Nilu Goonetilleke[1,5,6]*

[1]Department of Microbiology and Immunology, University of North Carolina, Chapel Hill, United States; [2]Lineberger Comprehensive Cancer Center, University of North Carolina, Chapel Hill, United States; [3]UNC Center For AIDS Research, University of North Carolina, Chapel Hill, United States; [4]British Columbia Centre for Excellence in HIV/AIDS, Vancouver, Canada; [5]Department of Medicine, University of North Carolina, Chapel Hill, United States; [6]UNC HIV Cure Center, University of North Carolina, Chapel Hill, United States; [7]Department of Urology, University of California San Francisco, San Francisco, United States; [8]Gladstone Institute of Virology and Immunology, San Francisco, United States; [9]Faculty of Health Sciences, Simon Fraser University, Burnaby, Canada

**Abstract** HIV-1-specific CD8+ T cells are an important component of HIV-1 curative strategies. Viral variants in the HIV-1 reservoir may limit the capacity of T cells to detect and clear virus-infected cells. We investigated the patterns of T cell escape variants in the replication-competent reservoir of 25 persons living with HIV-1 (PLWH) durably suppressed on antiretroviral therapy (ART). We identified all reactive T cell epitopes in the HIV-1 proteome for each participant and sequenced HIV-1 outgrowth viruses from resting CD4+ T cells. All non-synonymous mutations in reactive T cell epitopes were tested for their effect on the size of the T cell response, with a$\geq$50% loss defined as an escape mutation. The majority (68%) of T cell epitopes harbored no detectable escape mutations. These findings suggest that circulating T cells in PLWH on ART could contribute to control of rebound and could be targeted for boosting in curative strategies.

**\*For correspondence:**
nilu_goonetilleke@med.unc.edu

## Introduction

Antiretroviral therapy (ART) has transformed human immunodeficiency virus type 1 (HIV-1) from a fatal disease to a chronic condition. However, ART must be taken life-long as interruption of therapy in people living with HIV-1 (PLWH) mostly results in viral rebound within weeks (*Davey et al., 1999*; *Crowell et al., 2019*; *Bar et al., 2016*; *Buzon et al., 2014*). This rebound results from cells harboring HIV-1 DNA that is integrated into the host genome. While >95% of proviral DNA is replication-incompetent, the remaining fraction that we define here as the 'HIV-1 reservoir', retains the capacity to produce infectious virus particles, either stochastically (*Chun et al., 1997*; *Finzi, 1997*; *Siliciano et al., 2003*; *Eisele and Siliciano, 2012*), or through interventions to induce coordinated reactivation (*Archin et al., 2012*; *Søgaard et al., 2015*; *Elliott et al., 2013*; *Elliott et al., 2015*). The largest and best characterized HIV-1 reservoir resides in circulating resting CD4+ T cells. The inducible form of the HIV-1 reservoir is present in blood cells at a low frequency (~1 in $10^6$ resting CD4+ T cells), but is highly stable, with an estimated half-life of 44 months (*Crooks et al., 2015*; *Siliciano and Siliciano, 2015*; *Siliciano and Siliciano, 2004*). Natural decay of the HIV-1 reservoir

would therefore take >70 years (*Crooks et al., 2015*; *Siliciano and Siliciano, 2015*; *Siliciano and Siliciano, 2004*).

Strategies to allow PLWH to cease ART without viral rebound aim to either eliminate all HIV-1 infected cells harboring persistent, replication-competent virus (HIV-1 eradication), or to achieve a state of durable HIV-1 suppression without rebound (HIV-1 remission). Both approaches may harness HIV-1-specific CD8+ T cells to achieve reservoir reduction or elimination. T cells recognize infected cells through T cell receptors (TCR) that bind viral peptides presented on human leukocyte antigen (HLA) on the surface of the infected cell (*Townsend and McMichael, 1985*; *Townsend et al., 1984*; *Townsend et al., 1986*). On peptide recognition, CD8+ T cells can produce cytokines or release lytic molecules killing the infected cell. There is significant evidence that CD8+ T cells contribute to the control of untreated HIV-1, supported by studies of simian immunodeficiency virus (SIV) in non-human primates (*Pandrea et al., 2011*; *Barouch et al., 2002*; *Schmitz et al., 2005*; *McBrien et al., 2020*). First, certain HLA-I types, such as B*57:01/:03 and B*27:05, are overrepresented in HIV-1 elite virus controllers, defined as PLWH who exhibit sustained HIV-1 control below 50 copies/ml without ART. This suggests that specific subpopulations of CD8+ T cells are capable of controlling HIV-1 replication in the absence of therapy. Second, HIV-1-specific CD8+ T cells exert significant immune pressure on HIV-1, resulting in the accumulation of non-synonymous mutations in and around reactive T cell epitopes that can confer escape from CD8+ T cell recognition and killing (*Borrow et al., 1997*; *Goulder et al., 1997*; *Phillips et al., 1991*; *Price et al., 1997*). HIV-1 escape from CD8+ T cells, which is typically measured as a $\geq$ 50% decrease in the magnitude of the T cell response, can occur within days to weeks of infection (*Borrow et al., 1997*; *Price et al., 1997*; *Koup et al., 1994*; *Goonetilleke et al., 2009*). Both T cell immunodominance (relative strength of a T cell response) and viral entropy (sequence conservation of HIV-1 at a population level) are major determinants of the rate of virus escape in untreated infection (*Liu et al., 2013*; *Ferrari et al., 2011*; *Barton et al., 2016*). Notably, virus escape is ongoing in untreated HIV-1 infection (*Allen et al., 2005*), with new virus variants emerging over time in response to newly induced T cell responses, shifts in T cell clonotypes, or to compensate for fitness costs of earlier mutations. Lastly, when T cell selection pressure is removed, as in the case of virus transmission to recipients with HLA alleles different from the donor, reversion of mutations can occur (*Leslie et al., 2004*; *Carlson et al., 2014*).

ART is highly effective in inhibiting viral replication with multiple studies reporting that HIV-1 does not undergo directional selection during suppressive ART (*Kearney et al., 2014*; *Kearney et al., 2016*; *Rosenbloom et al., 2017*; *Van Zyl et al., 2017*). Recent studies have reported that the majority of proviruses, both total HIV-1 DNA and replication-competent viruses, that persist during ART, reflect viruses circulating during the year prior to ART initiation (*Brodin et al., 2016*; *Abrahams et al., 2019*; *Pankau et al., 2020*). These studies suggest that ART initiation is the major driver of latency formation, including the formation of the replication-competent stable HIV-1 reservoir. Proviruses that persist during ART would therefore include HIV-1 variants that have escaped from T cell selection over the entire course of untreated infection (*Phillips et al., 1991*; *Deng et al., 2015*). ART also impacts the HIV-1-specific T cell response, with reduced HIV-1 loads resulting in lower antigen drive and over time, yielding improved proliferation and a less exhausted T cell phenotype (*Rehr et al., 2008*; *Rutishauser et al., 2017*) (reviewed in *Warren et al., 2019*). We have recently shown that HIV-1-specific T cell responses are detectable ex vivo and are highly stable in PLWH on durable ART suppression (*Xu et al., 2019*), with others showing that ex vivo isolated CD8 + T cells can suppress HIV-1 superinfected CD4+ T cells as well as in vitro reactivated reservoir virus (*Sung et al., 2015*; *Shan et al., 2012*).

Collectively, these data suggest that HIV-1-specific T cell responses on ART should reflect those detected prior to the onset of ART. Likewise, the persistent HIV-1 reservoir should reflect the burden of virus escape mutants circulating pre-ART. Of note, a recent study reported that up to 98% of proviral HIV-1 sequences harbored mutations in immunoprevalent viral epitopes, though unmutated targets were also identified (*Deng et al., 2015*). A high burden of HIV-1 escape variants in the reservoir could limit approaches that aim to increase anti-HIV-1 immunity (*Ondondo et al., 2016*; *Borducchi et al., 2016*; *Mothe et al., 2015b*; *Achenbach et al., 2015*; *Hanke, 2014*).

Here, we present the first study reporting within-host HIV-1 escape frequencies in PLWH on ART. A non-selective approach was taken, specifically sequencing HIV-1 outgrowth viruses (OGVs) from the reservoir induced from resting CD4+ T cells, and mapping circulating T cell responses against

the entire HIV-1 proteome in 25 PLWH on ART. These strategies enabled us to empirically determine the extent and patterns of pre-ART escape within the stable HIV-1 reservoir. We observed that the majority, 68%, of T cell epitopes in the HIV-1 reservoir did not harbor escape variants and were recognized by circulating T cells. Furthermore, analysis of 'escape' epitopes (the remaining 32%) showed that escape was significantly lower in regions of HIV-1 that are more conserved. We conclude that the majority of circulating HIV-1-specific T cells in PLWH on ART can recognize the stable HIV-1 reservoir.

## Results

### Study cohort

HIV-1-specific T cell responses and the size of the HIV-1 reservoir were measured in 25 PLWH (ages 25–66, mean 46; male = 16, female = 9) durably suppressed by combination ART (viral load <50 copies/ml for 1.6–20.1 years, mean 7.2 years). Of the 25 participants, four initiated ART during acute HIV-1 infection, while the other 21 initiated ART during chronic HIV-1 infection (see Materials and Methods for definitions). Most participants (n = 24) were HLA-typed. Characteristics of the cohort are summarized in *Table 1* and full clinical details and IRB approvals are provided in Materials and Methods.

### HIV-1 outgrowth viruses in PLWH on ART show no directional selection over time

The HIV-1 reservoir in resting CD4+ T cells was measured using quantitative viral outgrowth assays (QVOAs) (*Lee et al., 2017*; *Siliciano and Siliciano, 2005*; *Trumble et al., 2017*; *Ho et al., 2013*; *Massanella and Richman, 2016*) for all 25 participants (*Figure 1*). Longitudinal IUPM measurements were obtained for 10 participants (*Supplementary file 1*). From these data, we calculated the mean of IUPM measurements (mean number of IUPM = 2/participant, range 1–6) made in the 2 years prior to HIV-1 T cell epitope mapping described below. Within the cohort, measurements ranged from below the limit of detection (participants 00941 and 20296, see Material and Methods for definition of limit of detection) to 2.528 IUPM (*Table 1*, *Supplementary file 1*).

OGVs were then sequenced. Primer ID sequencing (*Lee et al., 2017*; *Emery et al., 2017*), which allows for simultaneous in-depth sequencing of multiple HIV-1 regions was performed in one participant (participant 00673, mean of 3,700 Primer ID template consensus sequences/epitope). This approach was replaced with PacBio SMRT sequencing of the p24+ wells *at endpoint dilution* (i.e. average one virus-infected cell/well) to enable nearly, full-length sequencing of the HIV-1 genome. PacBio sequencing provided long reads (two reactions spanned almost the entire length of HIV-1) which identified sites of virus diversity within and outside epitopes that were reactive in initial T cell mapping (see Materials and Methods describing 'stripes'). Note that given the initial effective population size in each well was, on average, n = 1 (*Rouzine et al., 2001*), in vitro mutations occurred stochastically, with only 4–5 mutations expected across the virus genome over the 15 day QVOA culture (*Song et al., 2012*; *Brown and Richman, 1997*). Mutations arising in culture were therefore not expected to bias results.

Phylogenetic analysis of the 22 participants for whom near full-length sequencing was available (mean = 16 sequences per participant, range 1–71) confirmed that sequences from each participant formed monophyletic clades (*Figure 2A and B*). Participants were infected with HIV-1 clade B, except for one participant (participant 00926) who was infected with HIV-1 clade G. Available HIV-1 sequences from participants treated during acute infection (n = 3) showed greater genetic similarity (maximum pairwise diversity 0.0007) than sequences from participants treated during chronic infection (n = 19) (maximum pairwise diversity 0.0088) (Mann-Whitney two tailed test, p=0.003, *Supplementary file 2*). This is consistent with reports in which most HIV-1 infections are established from a single HIV-1 variant, and therefore the early onset of ART, which halts viral evolution, results in a smaller and more homogenous HIV-1 reservoir (*Kearney et al., 2014*; *Keele et al., 2008*; *Haaland et al., 2009*; *Derdeyn et al., 2004*; *Abrahams et al., 2009*).

For seven participants, OGVs from longitudinal samples were available for sequencing (up to 4.4 years, 2–4 time points/participant) (*Figure 3*). OGV sequences from different timepoints for each participant were interspersed within-host phylogenetic trees, suggesting no major directional

**Table 1.** Demographic and clinical data of participants (n = 25).

| Participant | Sex | Race[*] | Ethnicity[†] | Status at diagnosis[‡] | Age[§] | Years suppressed[¶] | Hiv-1 RNA[**] | Nadir CD4 | Mean IUPM [n][††] | HLA type |
|---|---|---|---|---|---|---|---|---|---|---|
| 00231 | M | C | NH | A | 66 | 9.66 | 1.5E+07 | 277 | 0.839 [1] | A*11:01, A*24:02, B*35:TDS[‡‡‡], B*35:02, C*04:01, C*04:01 |
| 00250 | M | C | NH | A | 48 | 7.44 | 6.8E+05 | 403 | 0.318 [3] | A*01:01, A*31:01, B*27:03, B*44:02, C*02:02, C*06:02 |
| 00531 | M | C | NH | C | 52 | 14.87 | 3.8E+05 | 81 | 1.372 [5] | A*02:01, A*02:01, B*07:02, B*44:02, C*03:04, C*07:02 |
| 00546 | M | C | NH | C | 56 | 20.09[§§] | N/A | 195 | 0.453 [2] | A*02:01, A*32:01, B*15:01, B*40:01, C*02:02, C*03:04 |
| 00673 | M | C | NH | C | 49 | 8.05 | 3.2E+05 | 168 | 0.790 [4] | A*24:02, A*30:02, B*15:17, B*18:01, C*05:01, C*07:01 |
| 00674 | M | C | NH | C | 57 | 6.25 | 6.8E+04 | 338 | 2.064 [5] | A*02:01, A*03:01, B*07:TDVB[§§§], B*40:01, C*03:04, C*07:02 |
| 00720 | M | C | NH | C | 25 | 2.96 | 4.0E+05 | 168 | 0.499 [4] | A*01:01, A*02:01, B*07:02, B*18:01, C*07:02, C*12:03 |
| 00728 | M | C | NH | C | 30 | 6.00 | 1.8E+04 | 354[¶¶] | 0.676 [6] | A*02:01, A*03:01, B*07:02, B*40:02, C*02:02, C*07:02 |
| 00749 | M | AA | NH | C | 25 | 4.10 | N/A | 402 | 0.365 [2] | A*03:01, A*30:02, B*15:03, B*40:01, C*02:10, C*03:04 |
| 00773 | M | C | NH | A | 46 | 2.01 | 2.4E+04 | 234 | 0.233 [1] | A*02:01, A*11:01, B*18:01, B*51:01, C*01:02, C*02:02 |
| 00833 | M | AA | NH | C | 36 | 8.16 | 7.9E+03 | 358[***] | 0.048 [1] | A*02:02, A*74:01, B*15:16, B*82:01, C*03:02, C*14:02 |
| 00834 | M | AA | NH | C | 25 | 2.15 | 2.1E+04 | 365 | 0.479 [1] | A*02:05, A*30:02, B*51:01, B*58:01, C*04:01, C*16:01 |
| 00848 | F | C | NH | C | 44 | 4.78 | 1.4E+04 | 800 | 0.022 [1] | A*01:01, A*30:02, B*08:01, B*18:01, C*05:01, C*07:01 |
| 00870 | M | C | NH | C | 54 | 5.28 | 5.3E+04 | 154[†††] | 1.400 [1] | A*11:01, A*24:02, B*35:01, B*55:01, C*03:03, C*04:01 |
| 00879 | F | AA | NH | C | 54 | 1.60 | 4.5E+02 | 612 | 0.093 [1] | A*03:01, A*33:01, B*14:02, B*58:02, C*06:02, C*08:02 |
| 00926 | F | AA | NH | C | 38 | 3.83 | 1.3E+05 | 508 | 0.262 [1] | A*02:02, A*02:05, B*:39:10, B*45:01, C*12:03, C*16:01 |
| 00929 | F | AA | NH | C | 61 | 1.77 | 1.9E+04 | 528 | 0.268 [1] | A*01:01, A*74:AB[****], B*15:MJMN[¶¶¶], B*81:AB[****], C*02:10, C*18:01 |
| 00930 | M | C | NH | C | 53 | 2.70 | 6.4E+05 | 174 | 2.084 [1] | A*01:01, A*11:01, B*08:01, B*40:01, C*03:04, C*07:01 |
| 00937 | F | AA | NH | C | 45 | 5.84 | 8.8E+04 | 260 | 1.276 [2] | A*03:01, A*30:01, B*35:01, B*52:01, C*04:01, C*16:01 |
| 00941 | F | C | NH | C | 47 | 14.58 | 6.6E+03 | 12 | 1.50E-08[‡‡][1] | N/A |
| 01014 | F | C | H | C | 52 | 19.85 | 1.2E+06 | 16 | 0.338 [1] | A*23:01, A*74:01, B*07:02, B*41:02, C*15:05, C*17:03 |
| 01086 | F | AA | NH | C | 30 | 5.81 | N/A | 320 | 0.0529 [1] | A*02:05, A*03:01, B*07:06, B*53:01, C*04:01, C*07:02 |
| 01095 | F | C | NH | C | 54 | 6.56 | 1.1E+05 | 88 | 2.528 [1] | A*03:01, A*31:01, B*40:01, B*44:02, C*03:04, C*05:01 |
| 10430 | M | C | NH | C | 59 | 9.47 | 8.1E+04 | 78 | 1.367 [5] | A*02:01, A*30:01, B*13:02, B*51:01, C*06:02, C*14:02 |
| 20296 | M | C | NH | A | 35 | 6.90 | 2.5E+04 | 427 | 1.50E-08[‡‡][1] | A*29:02, A*31:01, B*08:01, B*44:03, C*07:01, C*16:01 |
| | | | | Mean: | 46 | 7.22 | 8.6E+05 | 293 | 0.775 | |
| | | | | Range: | 25–66 | 1.6–20.10 | 4.5E+02-1.5E+07 | 12–800 | 1.50E-08–2.528 | |

[*] C: White/Caucasian, AA: Black/African American.

[†] NH: Non-Hispanic; H: Hispanic.

[‡] A: Initiation of ART during acute HIV-1 infection, prior to establishment of setpoint viremia; C: Initiating of ART during chronic HIV-1 infection.

[¶] Durable suppression defined as time from ≤ 40 copies/ml to date of T cell measurement.

[**] HIV-1 RNA at start of combination ART.

[††] IUPM (infectious units per million). Mean of IUPM measurements over 2 years from date of T cell mapping.

[‡‡] Below the limit of detection.

[§§] 00546 on monotherapy from 1985 to 1996, initiated combination ART in 1996. Years suppressed calculated from data of initiation of cART.

[¶¶] CD4 count 261 on 8/17/17 (after cell collection date).

[***] Lowest CD4 value, CD4 nadir before ART initiation is 567.

[†††] Lowest CD4 value, CD4 nadir before ART initiation is 157.

[‡‡‡] B*35:01/B*35:42.

[§§§] B*07:02/B*07:61/B*07:16.

[¶¶¶] B*15:03/B*15:103/B*15:220.

[****] B*81:01/B*81:02.

selection over time while participants remained suppressed on ART (*Figure 3*, *Figure 3—figure supplement 1*). Note, the single long-branched sequence in participant 10430's phylogeny met quality scores for sequencing and checks for hypermutations without evidence of mislabeling and was therefore retained. Overall, these longitudinal data are consistent with observations that ART effectively halts viral replication and subsequent evolution (*Kearney et al., 2014*; *Nettles, 2005*). Given the lack of directional selection, longitudinal sequences for each participant were pooled, increasing the mean sequencing depth of OGVs from 16 to 25 sequences (range: 1–71) per participant (*Figure 3—figure supplement 2*). In general, identical OGV sequences (see definition in Materials and Methods) were identified only infrequently; however, the 5' and 3' half genome sequences from two participants (participants 00250 and 00749, *Figure 3A and E*) exhibited >20% clonality, consistent with reports that in that the HIV-1 reservoir is in part maintained by clonal expansion (*Supplementary file 3*; *Bui et al., 2017*).

## The HIV-1 reservoir retains susceptibility to HLA-associated T cell responses

Bioinformatics approaches have been useful in identifying HIV-1 polymorphisms that are statistically significantly enriched among individuals expressing specific HLA class I alleles, identifying these as putative sites of viral escape (*Carlson et al., 2012*). Many of these HLA-associated polymorphisms have been experimentally verified as immune escape mutations, and lists of these HLA-associated polymorphisms are particularly comprehensive for HIV-1 subtype B (*Carlson et al., 2012*). We examined the OGV sequences recovered from 21 clade B-infected participants for whom both HIV-1 sequencing and HLA typing had been performed for the presence of known HLA-associated polymorphisms. For each HIV-1 sequence, each HLA-associated viral site (total number of HLA-associated sites n = 2,819, *Supplementary file 4*) was classified as either 'nonadapted', 'adapted', or 'possibly adapted'. 'Nonadapted' viral sites exhibited the specific HIV-1 residue predicted to be susceptible to the restricting HLA, 'adapted' sites exhibited the specific HIV-1 residue predicted to confer escape from the restricting HLA, and 'possibly adapted' sites exhibited any residue other than the 'nonadapted' form, supporting it as a possible escape variant. Minimum and maximum bounds of HLA-associated adaptation in each participant's reservoir were estimated by computing the total proportion of HLA-associated sites exhibiting 'adapted' forms (minimum bound), and those exhibiting either 'adapted' or 'possibly adapted' forms (maximum bound) (*Figure 4A*, *Supplementary file 4*; *Carlson et al., 2012*). Within participants, the proportion of HLA-associated sites exhibiting the HLA-adapted form ranged from 9–33% (adapted) and 32–66% (adapted + possibly adapted), and varied markedly by HIV-1 protein (*Carlson et al., 2012*). Importantly, our observation that, across all HIV-1 proteins, the minimum and maximum bounds for within-host escape burden ranged from 20% (adapted only) to 48% (adapted + possibly adapted) meant that between 52 and 80% of HLA-associated sites in the HIV-1 reservoir were predicted to retain susceptibility to host HLA-restricted T cell responses.

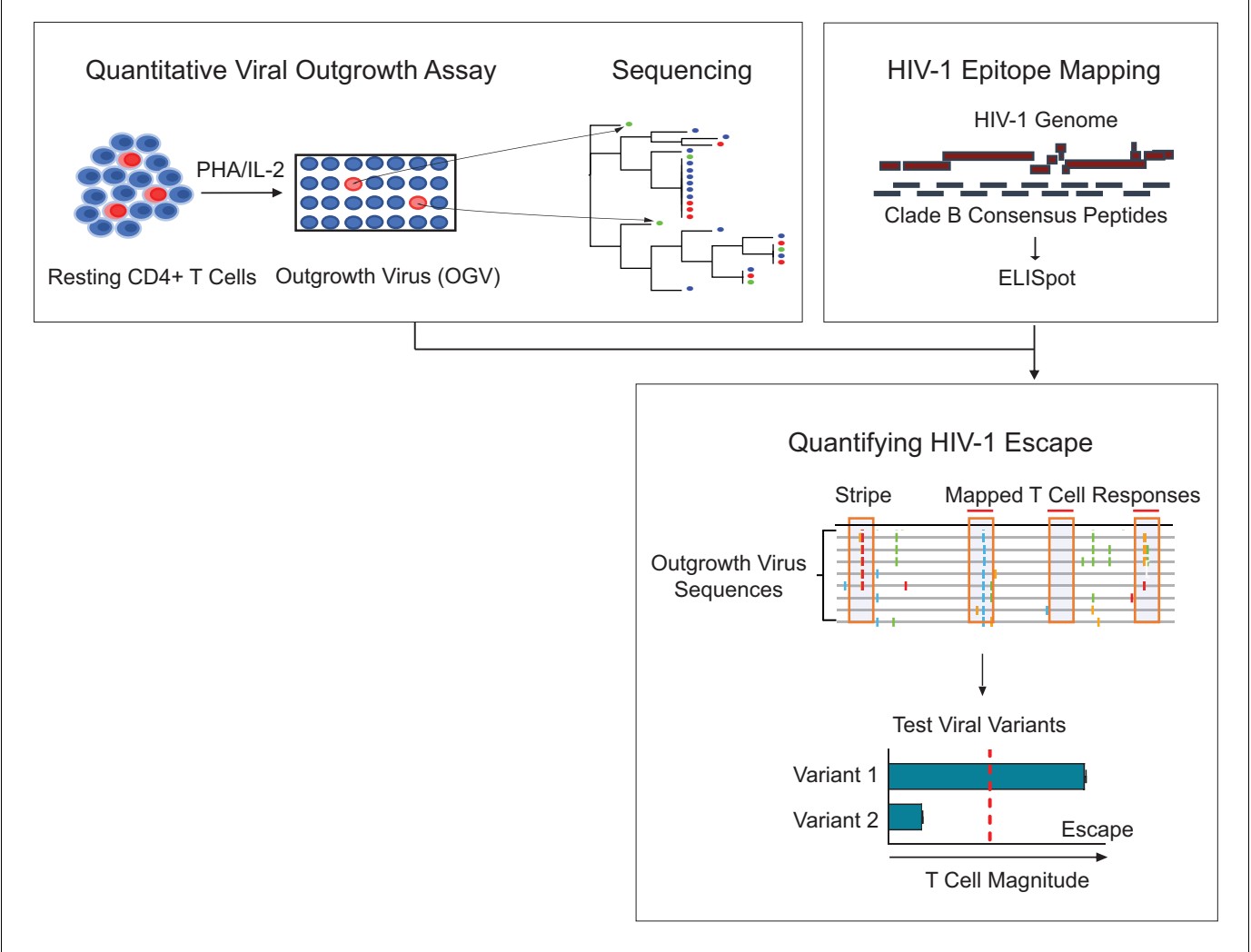

**Figure 1.** Schema of empirical testing of T cell escape in the HIV-1 reservoir. Top left panel. The quantitative viral outgrowth assay (QVOA) provided a minimal estimate of the size of the replication-competent reservoir virus from total resting CD4+ T cells of PLWH on ART (n = 25). Supernatants from HIV-1 p24+ wells harboring outgrowth virus (OGV) were sequenced by either PacBio (n = 22) or MiSeq (n = 1). Top right panel. Ex vivo IFN-γ ELISpot assays identified reactive T cell epitopes in clade B HIV-1 proteome in each participant. Lower panel. Sequencing and T cell data were combined to identify putative virus escape mutations by overlaying reactive T cell reactive epitopes (red horizontal line) on highlighter plots of each participant's outgrowth sequences (n = 23). Peptides spanning <u>all</u> non-synonymous changes between the clade B-defined epitope and outgrowth sequences (orange boxes) were synthesized. In addition, peptides matching OGVs in participants in which > 40% of viruses contained the same non-synonymous mutation generating a 'stripe' on the highlighter plot were also synthesized (orange box, no overhead red line). Variant peptides were examined for their impact on T cell recognition in ex vivo IFN-γ ELISpot. Virus escape was defined as a ≥ 50% difference in magnitude between variants, either Clade B sequence or OGV variants (vertical, red dotted line).

## HIV-1-specific T cell responses are detectable in PLWH on ART

To empirically assess the level of T cell escape, we mapped HIV-1-specific T cell responses in the 25 PLWH on ART. T cell responses were quantified by ex vivo IFN-γ ELISpot using a matrix method containing overlapping peptides spanning the consensus sequence of the entire clade B HIV-1 proteome (*Supplementary file 5*; *Goonetilleke et al., 2009*). While HIV-1-specific T cell responses remain detectable in PLWH on ART, the overall magnitude of the T cell response is ~10 fold lower than in untreated HIV-1 infection, which may result in the underestimation in the breadth of T cell responses (*Xu et al., 2019*; *Addo et al., 2003*). To increase ELISpot assay sensitivity, we added a higher number of PBMCs, $4 \times 10^5$ cells, to each well. After the initial matrix ELISpot, individual 18 amino acid (18-mer) peptides were tested to confirm reactive T cell responses. In cases in which overlapping 18-

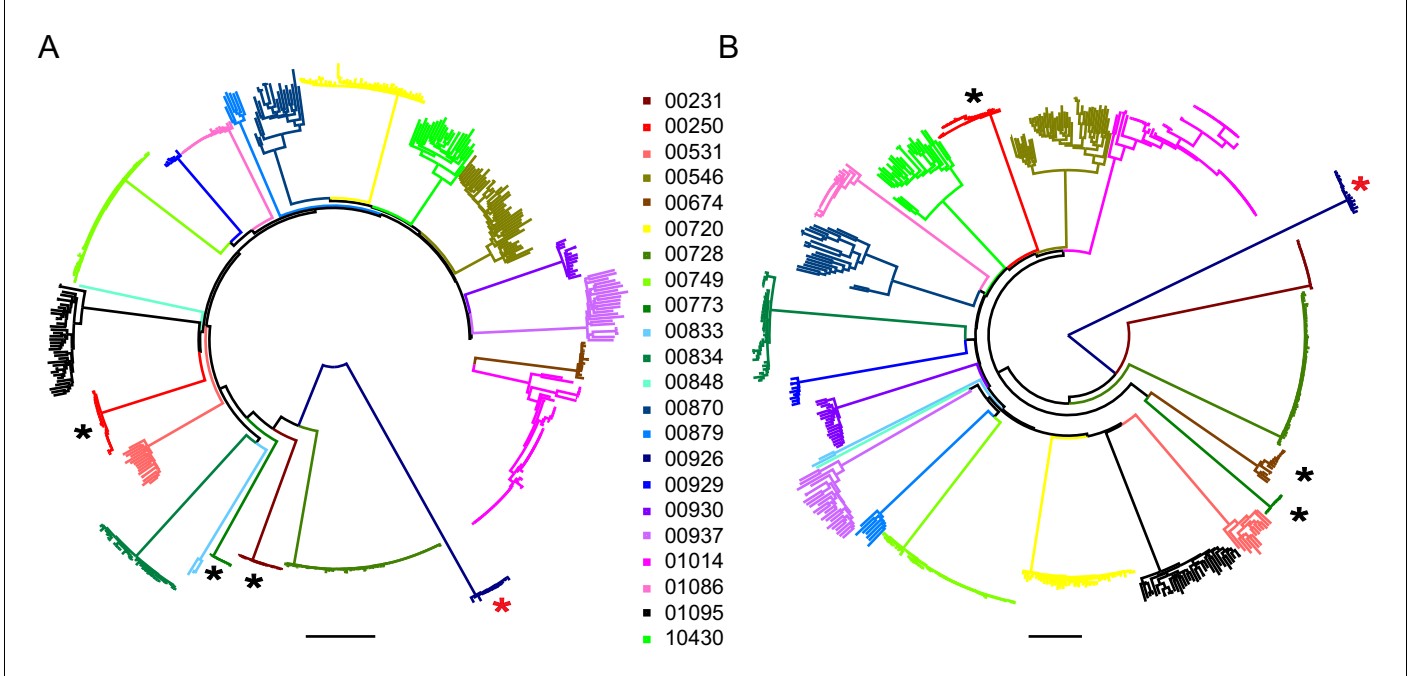

**Figure 2.** Neighbor-joining phylogenetic trees of sequenced outgrowth virus. The phylogenetic trees display 5' (**A**) and 3' (**B**) half genome sequences generated by PacBio sequencing of outgrowth virus from the quantitative viral outgrowth assay from n = 22 PLWH on ART. Each color indicates outgrowth virus sequences from a different PLWH on ART. Black stars denote participants who initiated ART during acute infection. All participants were infected with a clade B HIV-1 virus, except for one participant (red star), who was infected with a clade G HIV-1 virus. The genetic distance is shown by scale bar, 2 substitutions out of 100 bp. Trees are midpoint-rooted.

mer peptides were both reactive, we assumed responses were driven by a single reactive T cell epitope and only the higher magnitude response was designated reactive. Across the 25 participants, a total of 166 reactive T cell epitopes were identified (see Methods for positivity criteria). HLA typing was used to predict optimal CD8+ T cell epitopes, which were tested alongside 18-mer peptides. Optimal T cell epitopes, defined as inducing a T cell response ≥ than the corresponding 18-mer, were confirmed for 25/166 epitopes (*Supplementary files 5* and *6*).

The mean summed magnitude of HIV-1-specific T cell responses was 1,033 SFU/M (range: 156–2,856 SFU/M), and the mean T cell breadth was seven reactive epitopes (range: 1–19 epitopes) per participant (*Table 2*). Predictably, and consistent with reports in untreated HIV-1 infection, the magnitude of the mean summed HIV-1-specific T cell response correlated positively with the breadth of reactive epitopes targeted (r = 0.57, Spearman Rank two-tailed, p=0.003, 25 pairs, *Supplementary file 7*; *Addo et al., 2003*). As noted above, while the magnitude of the HIV-1-specific T cell response was weaker compared to untreated HIV-1 infection, the pattern of HIV-1 proteins targeted by T cells was similar to previous reports of untreated infection (*Figure 4B*; *Xu et al., 2019*; *Addo et al., 2003*). For example, the highest magnitude T cell response within a participant (immunodominant T cell response) most frequently targeted Gag and Nef, whereas the most frequently targeted protein across the cohort (immunoprevalent T cell response) was Gag (*Figure 4—figure supplement 1*; *Addo et al., 2003*). No differences in either T cell magnitude or breadth were observed between women and men, or between participants treated during acute versus chronic HIV-1 infection (*Supplementary file 7*). Neither T cell magnitude nor breadth exhibited any association with measured HIV-1 IUPM (*Supplementary file 8*; *Xu et al., 2019*).

## Escape is not observed in the majority of T cell epitopes in PLWH on ART

To empirically quantify T cell escape in this cohort, OGV variants were tested for their effect on T cell response (*Figure 4C*). In participants for which > 1 OGV sequences were obtained (n = 23 participants and collectively 151 epitopes), peptide/s were synthesized to match each participant's OGV

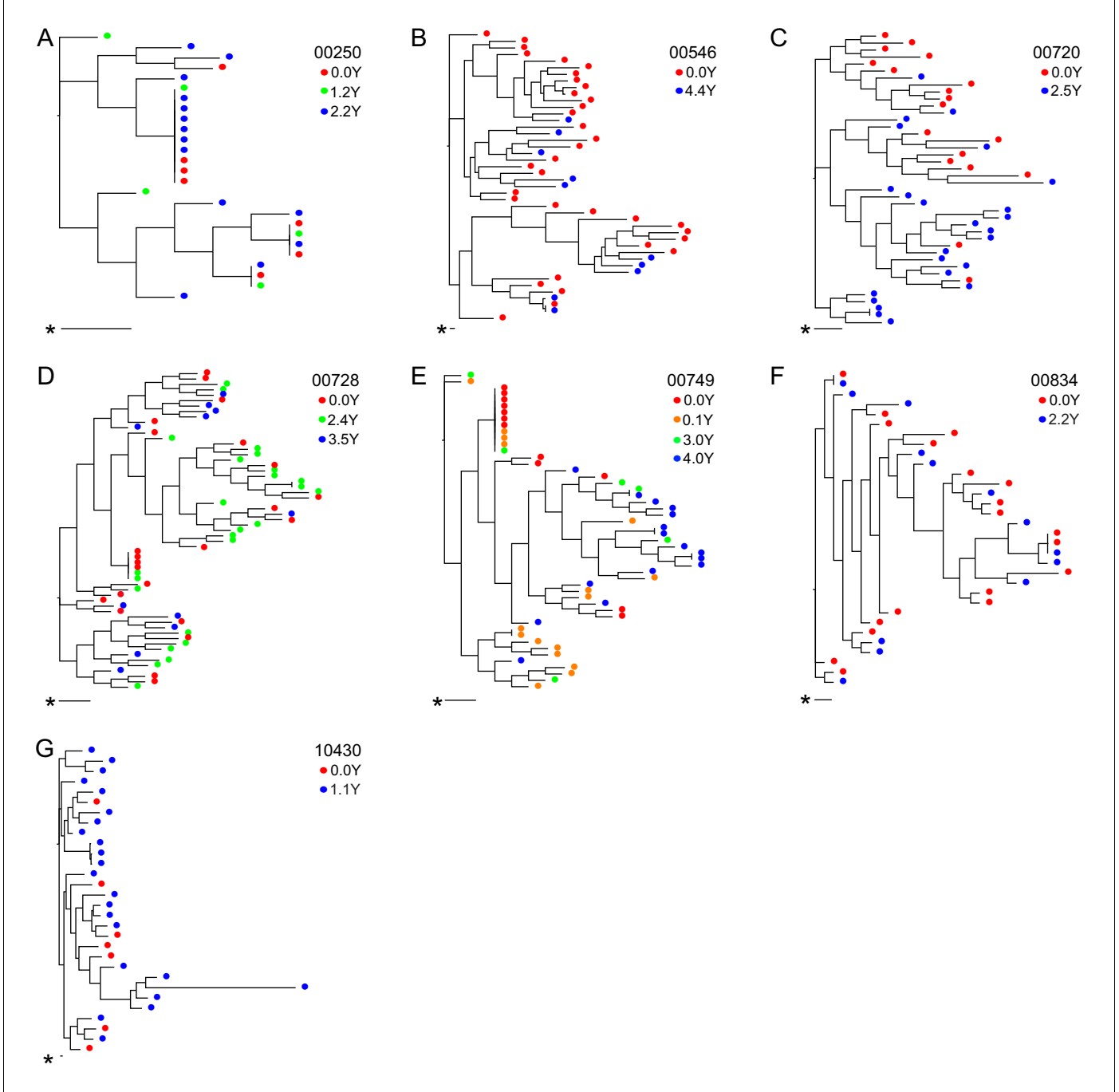

**Figure 3.** Neighbor-joining within-host phylogenetic trees inferred from longitudinal outgrowth virus sequences. Phylogenetic trees displaying 3' half genome sequences of longitudinal outgrowth virus sequences from seven participants (A–G). Each color represents a different time point in which outgrowth virus was sequenced for each participant. Sequences from different time points for each participant are distributed throughout the phylogeny. The genetic distance is shown by scale bar (indicated with an asterisk), 1 substitution out of 1000 bp. Trees are midpoint-rooted. (A) 00250 treated during acute infection and (E) 00749 have > 20% clonal sequences.

The online version of this article includes the following figure supplement(s) for figure 3:

**Figure supplement 1.** Longitudinal outgrowth virus sequences on neighbor-joining phylogenetic trees.

**Figure supplement 2.** Size of the replication-competent reservoir as measured by infectious units per million (IUPM) against the number of sequences obtained by PacBio sequencing (r = 0.755, p<0.0001, Spearman Rank, n = 24 pairs).

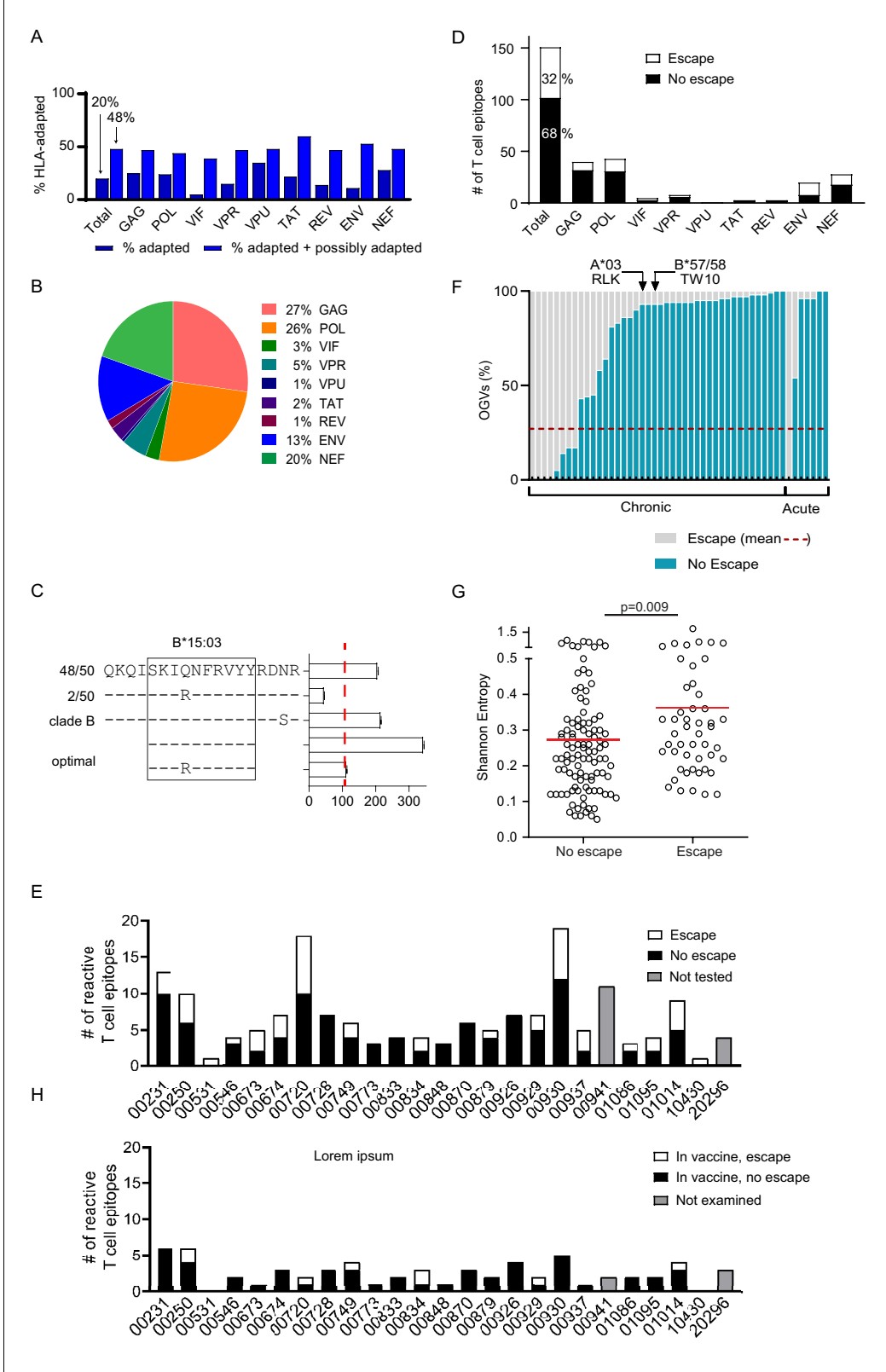

**Figure 4.** Quantifying T cell escape in the HIV-1 reservoir. (**A**) *HLA associations in OGVs*: The proportion of polymorphisms harboring HLA-adapted (blue bar) and HLA-adapted + possibly HLA-adapted polymorphisms (light purple) in PLWH on ART (Clade B infected with full-length sequencing, n = 22). Results are summarized as total, as well as by individual HIV-1 protein. (**B**) *T cell mapping*: Ex vivo IFN-γ ELISpot was performed to determine T cell responses (n = 166), shown by protein, against the clade B HIV-1 proteome in 25 PLWH on ART. (**C**) *Escape testing*: Variants detected in OGVs of
*Figure 4 continued on next page*

*Figure 4 continued*

participant 00749 were tested ex vivo using IFN-γ ELISpot alongside the original reactive clade B 18-mer and optimal peptide. Bars show the T cell magnitude reported as spot forming units per million (SFU/M) PBMCs. Escape was defined as a ≥ 50% decrease (red vertical dotted line) in the mean magnitude of triplicate wells compared to the peptide which elicited the greatest T cell response. The left side shows the proportion of each variant in total OGVs (n = 50). (**D**) *Cohort-level escape*: T cell escape was examined in participants for which OGV sequence data were available (n = 23 participants, n = 151 epitopes). The proportion of empirically confirmed T cell epitopes that harbored escape variant(s) (n = 49, white) or no escape variant(s) (n = 102, black) in OGV. Results are summarized in total, as well as by individual HIV-1 protein. (**E**) *Escape by participant*: The proportion of reactive T cell epitopes that harbored escape variant(s) (white) and no escape variant(s) (black) per participant (n = 23). Gray bars indicate participants (n = 2) where epitopes (n = 15) were not tested for escape due to limited OGVs. (**F**) *Within-epitope escape*: Stacked bar graph showing the proportion of individual OGVs that conferred (gray –escape) or did not confer (blue – no escape) T cell escape in each epitope for which escape was observed (n = 49). Red dotted line is the average (27%) frequency of OGVs that conferred T cell escape across all epitopes. Data are ordered in ascending order of no escape OGVs, grouped by whether the participants initiated ART in either acute or chronic HIV-1 infection. Arrows identify immunoprevalent T cell epitopes, in which the majority of OGVs do not harbor escape variants. (**G**) *Shannon entropy*: A Shannon entropy score was calculated for reactive T cell epitopes for which sequence data were available (n = 151, 23 participants). Entropy was higher in epitopes which contained T cell escape variant(s) (n = 49, median = 0.29) than epitopes with no escape variant(s) (n = 102, median = 0.23) in the OGVs (Mann Whitney two-tailed test, p=0.009). (**H**) *Escape in vaccine immunogen*: The proportion of reactive T cell epitopes in each participant (n = 25) that are targeted by the HIVconsvX immunogen (total = 59) and harbored one or more escape variants (white), or no escape variants (black) in OGVs. Gray bars indicate participants (n = 2) where epitopes in the immunogen could not be examined for escape due to limited OGVs.

The online version of this article includes the following figure supplement(s) for figure 4:

**Figure supplement 1.** HIV-1-specific T cell responses in PLWH on ART.

**Figure supplement 2.** Outgrowth virus sequences and DNA sequences on neighbor-joining phylogenetic trees.

autologous virus sequence/s. This included any viral variants that differed from the clade B peptide used to originally map the respective reactive T cell epitope. In total, 115 of 151 epitopes contained variant OGVs.

All variant peptides were tested ex vivo in an IFN-γ ELISpot assay alongside the originally tested 18mer clade B consensus peptide, and if identified, the optimal epitopes. Escape was defined as a ≥ 50% decrease in the average magnitude of triplicate wells compared to the peptide that elicited the greatest T cell response, whether autologous peptide or clade B consensus peptide (*Figure 4C*, *Supplementary file 6*). We would note that a range of escape thresholds, ranging from 50–80% (*Borrow et al., 1997*; *Goonetilleke et al., 2009*; *Radebe et al., 2015*), have been applied in previous HIV-1 studies. In this study, our goal was to maximize sensitivity to detect escape by identifying all outgrowth sequences in our dataset with the potential to compromise T cell immunity; therefore, the 50% threshold was applied.

Most variant peptides did not impact T cell magnitude. In total, virus escape was confirmed in the replication-competent HIV-1 reservoir in 49 of 151 (32%) HIV-1 T cell epitopes (*Figure 4D*) with escape occurring more frequently in epitopes in Pol, Env and Nef (25%, 23%, 21%) proteins (*Figure 4D*).

As described above, participants recognized an average of 7 T cell epitopes, ranging from 1 to 19. In each participant, one third (average = 34%) of reactive epitopes harbored virus escape mutations (*Figure 4E*, *Table 2*). In six participants, no T cell escape was observed in any reactive T cell epitope (i.e. 0% escape in 3–7 epitopes, additional DNA sequencing in *Figure 4—figure supplement 2*), whereas in the two participants who responded to only one epitope, virus escape was confirmed in both cases (i.e. 100% escape) (*Figure 4E*, *Table 2*).

## At each epitope, only a minority of outgrowth viruses confer T cell escape

We next examined each of the 49 epitopes in which escape was observed to quantify the fraction of sequenced OGVs that conferred escape that is the 'depth' of escape (*Figure 4F*). On average for each epitope, 27% of OGV sequences tested produced a ≥ 50% loss in T cell response (*Figure 4F*, *Supplementary file 6*). The range of within-epitope escape was broad, starting at 1–2% to 100% of all OGVs. Note, that an escape depth of 100% (which was observed for n = 5 epitopes) occurred when all OGVs were identical but induced a lower T cell response than the originally tested consensus clade B epitope. In these cases, we assumed that the primary T cell response was induced against a consensus epitope no longer detectable in the reservoir. No differences in the depth of

Table 2. HIV-1-Specific T cell reponses (n = 25 participants).

| Participant | Summed magnitude ($10^6$/PMBC) | Immunodominant T cell response | Total T cell breadth | # of epitopes with escape variant(s) in outgrowth virus | Escape in immunodominant epitope (Y/N) |
|---|---|---|---|---|---|
| 00231 | 2856 | Vif153→170 | 13 | 3 | Y |
| 00250 | 1531 | Env765→782 | 10 | 4 | Y |
| 00531 | 390 | Gag357→374 | 1 | 1 | Y |
| 00546 | 895 | Gag260→277 | 4 | 1 | N |
| 00673 | 2313 | Nef73→90 | 5 | 3 | N |
| 00674 | 1394 | Env838→855 | 7 | 3 | Y |
| 00720 | 1947 | Pol521→538 | 18 | 8 | Y |
| 00728 | 439 | Gag204→221 | 7 | 0 | N |
| 00749 | 418 | Pol481→498 | 6 | 3 | N |
| 00773 | 1174 | Pol273→290 | 3 | 0 | N |
| 00833 | 333 | Gag164→181 | 4 | 0 | N |
| 00834 | 725 | Pol273→290 | 4 | 2 | N |
| 00848 | 276 | Nef89→106 | 3 | 0 | N |
| 00870 | 363 | Nef65→82 | 6 | 0 | N |
| 00879 | 500 | Gag292→309 | 5 | 1 | N |
| 00926 | 234 | Gag180→197 | 7 | 0 | N |
| 00929 | 798 | Gag180→197 | 7 | 2 | N |
| 00930 | 2182 | Nef81→98 | 19 | 7 | Y |
| 00937 | 297 | Pol257→274 | 5 | 3 | N |
| 00941 | 2216 | Nef81→98 | 11 | N/A[*] | N/A[*] |
| 01014 | 1463 | Nef65→82 | 9 | 4 | N |
| 01086 | 258 | Gag17→34 | 3 | 1 | Y |
| 01095 | 2337 | Pol313→330 | 4 | 2 | N |
| 10430 | 157 | Nef97→114 | 1 | 1 | Y |
| 20296 | 331 | Gag204→221 | 4 | N/A[*] | N/A[*] |
| Total | | | 166 | 49 (32% - 49/151) | |
| Mean: | 1033 | | 7 | 2 | |
| Range: | 157–2856 | | 1–19 | 0–8 | |

[*]No outgrowth virus sequence data available to test for the presence of an escape variant in the sequenced outgrowth virus.

virus escape was observed between participants treated in acute or chronic infection (*Figure 4F*, Mann-Whitney two-tailed test, p=0.216). Our mapping included multiple, well-characterized immunoprevalent epitopes that also harbored a mix of escape and non-escape OGVs, but overall, mostly non-escape (*Figure 4F*, *Supplementary file 5*).

## Epitope entropy but not T cell immunodominance is associated with escape in PLWH on ART

We hypothesized that escape, as observed in studies of untreated HIV-1 infection, would be associated with less conserved regions of HIV-1. Shannon entropy is used to measure the level of HIV-1 population diversity at each position along the HIV-1 proteome (*Shannon, 1948*). Here, epitope entropy was calculated as previously described (*Liu et al., 2013*), as the mean entropy of each position within the epitope. High entropy values correspond to sites with more population diversity, that is sites that are less conserved (*Allen et al., 2005*; *Shannon, 1948*). As expected, escape was more

commonly observed in regions of higher Shannon entropy (Mann-Whitney two-tailed test, p=0.009) (*Figure 4G*).

While our previous work in untreated HIV-1 infection reported that immunodominance is a major determinant in the time to viral escape, immunodominance of circulating T cell responses in PLWH on ART did not associate with the presence or absence of escape within the cognate epitope (Mann-Whitney two-tailed test, p=0.971, *Figure 4—figure supplement 1*). This may be because effective ART in this study cohort prevented directional selection and immunodominance hierarchies subsequent to the initial escape event may have shifted, whether in untreated infection or at the time of ART initiation.

## Conserved HIV-1 vaccine immunogens preferentially target non-escaped T cell epitopes

Our and others' observations that T cell escape occurs more frequently in higher entropy (less conserved) epitopes (*Liu et al., 2013*; *Ferrari et al., 2011*; *Barton et al., 2016*) suggests that vaccine-induced HIV-1-specific T cells targeting low entropy (more conserved) regions may contribute to better HIV-1 control. This may be due to HIV-1-specific T cells maintaining selection pressure on regions of the virus where escape is likely to incur a fitness cost. Several conserved-element vaccine immunogens have been designed which connect the most conserved parts of the viral proteome, either directly in-frame or with linker sequences to form artificial conserved element immunogens. Vaccination with these immunogens aims to focus T cell responses against highly conserved epitopes that are generally subdominant in natural infection (*Barouch and Korber, 2010*; *Rolland et al., 2007*; *Mothe et al., 2015a*; *Borthwick et al., 2017*; *Gorse et al., 2008*). We interrogated four different conserved vaccine immunogen designs, HIVconsv (*Létourneau et al., 2007*), HIVconsvX (*Ondondo et al., 2016*), HIVACAT T cell Immunogen (HTI) (*Mothe et al., 2015a*), and the p24 conserved element (p24 CE) (*Kulkarni et al., 2013*), using our dataset to examine (1) whether the immunogen contained epitopes targeted by circulating T cell responses in PLWH on ART and (2) whether the epitopes in the immunogen contained escape variant(s) found in OGVs recovered from of our participants.

The 151 epitopes that were targeted by T cells in our study participants were differentially covered by these four immunogens. The HIVconsX immunogen contained the largest number of circulating T cell responses observed in our dataset (59/151; 39%; mean of 3 epitopes/participant, range 0–6) while the p24 CE vaccine contained the least (16/151; 11%; mean of 1 epitope/participant, range 0–4) (*Figure 4H*, *Supplementary file 9*). Nevertheless, escape frequency was on average two-fold lower in reactive T cell epitopes (12–29%) found within the immunogens relative to those excluded from immunogen designs (33–44%). For example, the overall proportion of reactive T cell epitopes within the HIVconsX immunogen that contained escape variants was 15.3% (9/59 epitopes, average 0 escape epitopes/participant, range 0–2), whereas the proportion of reactive T cell epitopes outside of the HIVconsX immunogen that contained escape variants was almost >2.5-fold higher (43.5%, average two escape epitopes/participant). This difference was statistically significant (p=0.001, Pearson's $\chi^2$ test with Yates correction, *Supplementary file 9*). Similarly, the other three conserved immunogen vaccines also targeted T cell epitopes with a low rate of escape (*Létourneau et al., 2007*; *Mothe et al., 2015a*; *Kulkarni et al., 2013*). This suggests that current conserved immunogen designs are likely to be immunogenic in PLWH on ART and are more likely to target T cell epitopes that do not harbor pre-ART escape.

## Discussion

In this study, we employed two unbiased approaches to comprehensively examine the landscape of escape in the HIV-1 reservoir in 25 PLWH durably suppressed on ART. First, we examined the frequency of published HLA-associated polymorphisms in HIV-1 subtype B OGV sequences. Overall, 20% of HLA-associated sites in the HIV-1 reservoir harbored the known HLA-adapted viral form, suggesting potential viral escape from T cell responses. This number increased to 48% if both known and putative HLA-adapted forms were included in the calculation. These estimates are consistent with substantial immune escape in the reservoir, but nevertheless support the notion that the majority of HLA-associated sites in the reservoir maintain susceptibility to host T cell responses. Next, in all participants, we mapped HIV-1-specific T cell epitopes across the full HIV-1 proteome. Across the

cohort, the mean breadth of HIV-1 epitopes targeted per person was 7. T cell targeting of HIV-1 proteins was consistent with studies of untreated infection, with T cells most often targeting Gag, Pol, and Nef (*Addo et al., 2003*). No differences were observed between treatment initiated in acute (n = 3) or chronic infection (n = 22). We then systematically examined whether non-synonymous mutations in reactive epitopes impacted T cell recognition and functional responses. Consistent with the overall HLA-adaptation estimates from the bioinformatics analysis, we experimentally confirmed that 32% (49/151) of all reactive T cell epitopes in HIV-1 OGVs contained mutations that conferred T cell escape, and that conversely, 68% of circulating HIV-specific T cell responses were able to fully recognize the autologous replication-competent-reservoir virus.

The minimal empirically defined T cell escape frequency of 32% observed in PLWH on ART in this study is consistent with studies of T cell escape in untreated HIV-1 infection. A 30% escape frequency was observed in primary untreated HIV-1 infection, in which HIV-1-specific T cell responses were comprehensively mapped against the autologous transmitted/founder virus in participants, and T cell reactivity was assessed against mutant viruses that emerged over the first 6 months of infection (*Liu et al., 2013*). In that study, similar to ours, CD8+ T cell escape occurred more frequently in epitopes of higher Shannon entropy (*Liu et al., 2013*; *Ganusov et al., 2011*; *Bansal et al., 2005*). A similar escape frequency of 33% was also observed in a large cross-sectional study of chronically infected HIV-1 Clade C untreated individuals (n >2,000), in which the T cell escape frequency was determined by examining participant HLA-associated polymorphisms in immunoprevalent HIV-1 CD8+ T cell epitopes (*Carlson et al., 2012*). In PLWH durably suppressed on ART, HIV-1-specific T cell responses are stable and maintain immunodominance hierarchies over years (*Xu et al., 2019*). Both the stability of the HIV-1-specific T cell response and the comparable level of population-level virus escape in treated and untreated individuals is consistent with studies of HIV-1 reservoir formation and kinetics. These studies showed that the HIV-1 reservoir is mostly formed at the time of ART initiation (*Brodin et al., 2016*; *Abrahams et al., 2019*; *Pankau et al., 2020*) and its subsequent rate of decay is very slow (*Crooks et al., 2015*; *Siliciano and Siliciano, 2015*; *Siliciano and Siliciano, 2004*), with only limited variation in the number of proviruses, or the types of proviral defects observed in PLWH on ART over time (*Lu et al., 2018*). More recently, longitudinal analyses of near full-length proviral sequences in PLWH on ART showed no enrichment in escaped epitopes over time suggesting that HIV-1-specific T cells do not significantly alter the provirus landscape in people durably suppressed (*Antar et al., 2020*). Furthermore, a recent study reported that brief analytical treatment interruption (ATI) (median of 5 weeks) does not significantly impact the composition of virus populations (phylogenetic analyses sequencing of HIV-1 env) or size (total DNA, cell-associated RNA, IUPM) of the latent HIV-1 reservoir (*Salantes et al., 2018*). Collectively, these studies suggest a model in which circulating T cells, as well as virus escape variants in infected cells at the time of ART initiation, are maintained throughout durable viral suppression, and during brief periods of ATI.

Our observations contrast somewhat with those of Deng and colleagues who reported a predominance of CD8+ T cell-resistant viruses in proviral HIV-1 DNA (*Deng et al., 2015*). Deng et al did not empirically investigate proteome-wide patterns of escape (i.e. escape in 32% of reactive T cell epitopes) in the replication-competent HIV-1 reservoir, as reported here. Rather, their study focused on predicting the 'proportion' of escape within epitopes in Gag. They mostly used sequencing and bioinformatics approaches to conclude that, in epitopes in which virus variation was observed, in individuals treated in chronic infection, most (>98%) proviruses contained variants previously reported to confer T cell escape. Our empirical testing, which focused on T cell escape in the persistent HIV-1 reservoir, did not yield the same high proportion of escape variants reported in that study. Indeed, the average within-epitope escape frequency in 49 empirically tested epitopes was much lower at 27% (*Supplementary file 5*). This dataset included escape variants that can emerge early after infection such as the Gag T242N escape in Gag 240–249, restricted by B*58:01 and B57:01/3 (*Leslie et al., 2004*; *Brumme et al., 2009*). Deng et al also noted a higher within-epitope escape in the DNA reservoir in individuals treated in chronic versus acute infection. While only a few participants in this study were treated in acute infection, we observed no difference in within-epitope escape frequencies between participants treated in chronic or acute HIV-1 infection. Together this suggests that the differences in the studies did not arise from the regions of HIV-1 studied, or the differences in stage of HIV-1 infection at time of ART initiation. Rather, we propose that differences in within-epitope escape frequencies between studies derived from two sources. First, our empirical approach showed clearly that HIV-1 variants often do not escape T cell recognition and therefore

prediction methods may overestimate the level of escape. Indeed, Deng et al did include examples where virus variants predicted to confer escape did not decrease T cell responses following experimental testing. Second, the HIV template, HIV DNA versus OGV RNA, used for virus sequencing largely differed between studies.

Our observations that epitopes mostly harbor a mix of escape and non-escape variants are consistent with previous studies in untreated infection reporting that T cell escape is rarely all-or-nothing, but rather occurs at different rates (days to years) with multiple variants emerging over time (*Goonetilleke et al., 2009*; *Liu et al., 2013*; *Asquith et al., 2006*). These changes reflect both ongoing compensatory changes to maintain viral fitness and selection pressure from different T cell clonotypes (*Brockman et al., 2010*; *Ladell et al., 2013*; *Ritchie et al., 2014*), and progressive dysregulation of the immune response. While the HIV-1 reservoir is mostly formed at ART initiation, almost 30% of sequences enter the HIV-1 reservoir much earlier. The mixture of escape and non-escape OGVs observed in this study likely represent OGVs at different stages of T cell escape entering the reservoir at different stages of infection.

There are limitations of this study. First, detection of escape variants may have been limited by the sequencing depth obtained from OGVs from QVOAs. In addition, different stimuli can activate different viruses, and even the negative wells from the QVOA can generate new viruses after new stimulation, contributing to an underestimation of, or a bias in the replication-competent viruses present in the culture (*Ho et al., 2013*; *Hosmane et al., 2017*). Applying a rule of three calculation, our average depth of sequencing (25 sequences/participant) provided 95% confidence that any undetected variant comprised <12% of the population. While sampling biases were minimized by our choice of sequencing methods (nearly full-length and Primer ID), which have a low probability of introducing PCR mutations (*Zhou et al., 2015*; *Salazar-Gonzalez et al., 2008*), we cannot exclude that a low-frequency variants not detected by sequencing may have conferred escape (*Abrahams et al., 2019*). It is notable that that participant OGVs sequenced over time did not exhibit directional selection suggesting that the QVOA method recovers outgrowth virus that is representative of all variants (escaped and non-escaped) in the HIV-1 reservoir. Second, our bioinformatics analysis of HLA-associated polymorphisms in HIV-1 reservoir sequences simply identifies viral variants that are enriched among individuals expressing a particular HLA allele, as defined using population-level analyses (*Carlson et al., 2012*), and thus provides only a crude estimate of adaptation burden. This is because these lists of HLA-associated polymorphisms do not capture every escape variant that occurs in vivo; moreover, not every HLA-associated viral site defined at the population level is necessarily targeted by every individual expressing the relevant HLA allele. It is notable, however, that the results of the bioinformatics approach were consistent with our empirical studies, suggesting that the estimates made correctly captured the range of escape frequencies within this cohort. Third, in our empirical studies, we cannot exclude that the use of clade B consensus peptides underestimated the total breadth of the T cell response (*Goonetilleke et al., 2009*; *Altfeld et al., 2003*). However, our approach of synthesizing autologous peptides at reactive epitopes, and experimentally testing all 'stripes' (non-synonymous changes in >40% of sequences, see Materials and Methods) in our OGVs to identify additional epitopes to study, minimizes this limitation. Fourth, our empirical approach did not detect all mechanisms of escape such as possible immune dysfunction and processing mutations. To fully assess loss of CD8+ T cell responses due to immune dysfunction, longitudinal mapping and sequencing studies pre -and post-ART are needed. While no previously documented processing mutations were detected in our dataset, we anticipate the bioinformatics analysis which enumerated all HLA-associated polymorphisms regardless of their location with respect to known (or novel) T cell epitopes, will have captured some processing mutations. Lastly, we did not confirm the CD4 or CD8 restriction of all reactive and escaped T cell epitopes in this study. However, both consistent with previous studies (*McNeil et al., 2001*; *Appay et al., 2002*; *Betts et al., 2001*; *Riou et al., 2012*) and the optimal epitope mapping performed here, it is likely that the bulk of T cell responses identified in this study were CD8-restricted.

In summary, empirical testing supported by bioinformatics analysis, showed that 68% of circulating HIV-1 specific T cells across our cohort were able to recognize the replication-competent virus from the latent reservoir. Moreover, T cells in most individuals targeted one or more epitopes with no detectable escape and within epitopes, on average, only a minority of OGVs conferred escape. Treatment interruption studies have shown that circulating HIV-1 specific T cell responses are insufficient to fully suppress virus rebound in most individuals. Recent work has shown recombinant viruses

rapidly emerge following HIV rebound (*Vibholm et al., 2019*; *Lu et al., 2018*; *Cohen et al., 2018*). Our study, supported by animal models (*Pandrea et al., 2011*) and case reports investigating viral rebound (*Smith et al., 2015*), suggest that circulating HIV-1-specific T cells very likely afford some contribution in control of virus rebound. However, T cell immunity is rapidly overwhelmed, in part due to recombination-mediated escape (*Ritchie et al., 2014*; *Streeck et al., 2008*). Altogether, this suggests that HIV-1 cure strategies that include efforts to augment T cell immunity in PLWH on ART should include immunogen designs that target existing T cell epitopes, particularly those with low entropy which, as we show here, are less likely to harbor escape in the replication-competent reservoir. Ultimately, success of these therapies will likely be dependent on the ability to exert very rapid virus control and to limit the emergence of viral recombinants.

## Materials and methods

### Study cohort

Participants were enrolled through either the UNC Chapel Hill HIV-1 Clinical Trials Unit, the Women's Interagency HIV-1 Study (WIHS) UNC Chapel Hill Site, or the University of California San Francisco (UCSF) site. All participants were adults (>18 years of age) with documented HIV-1 infection. Infection was confirmed by either a positive HIV test by licensed ELISA test and confirmed by western Blot (WB), multispot HIV assay, HIV-1 RNA, or two documented plasma HIV-1 RNA >1000 copies/ml.

Enrollment criteria included a stable ART regimen for $\geq$12 months prior to consent/enrollment. Stable ART was defined as $\geq$nucleoside/nucleotide reverse transcriptase inhibitors plus a non-nucleoside reverse transcriptase inhibitor, integrase inhibitors, or a protease inhibitor without interruption (missing doses for no more than 2 consecutive days or no more than 4 cumulative days) in the 24 weeks immediately prior to consent/enrollment. Other potent fully suppressive ART combinations were allowed on a case-by-case basis, after clinician review. Prior changes in or elimination of medicines for easier dosing, intolerance, or other reasons were permitted if an alternative suppressive regimen was maintained. Second, plasma HIV-1 RNA was required to be <50 copies/mL for $\geq$12 months prior to enrolment as defined by a minimum of two tests (limit of detection determined by assay employed). Third, within 60 days of consent/enrollment, plasma HIV-1 RNA was required to be <50 copies/ml and hemoglobin required to be $\geq$12 g/L. Treatment during acute HIV-1 infection was defined as treatment within 30 days of either HIV-1 PCR positive but HIV-1 antigen negative, or a HIV-1 antigen discordant test.

Class I HLA typing was performed on 24/25 participants the UNC Center for AIDS Research Core Laboratory. Study characteristics of the cohort are summarized in *Table 1*.

### Study approval/IRB approvals

Participants were enrolled with the following IRB approved studies: (1) *CID 0819 - Apheresis Procedures to Obtain Leukocytes for Research Studies from HIV Positive Participants (08–1575)*, (2) *The UNC Women's Interagency HIV Study (WIHS) (12-1660)*. Review and implementation of all protocols utilized for the collection of samples for this this analysis were approved by the University of North Carolina at Chapel Hill Biomedical Institutional Review Board (IRB) *and the University of California at San Francisco IRB*. All participants provided written informed consent. All experimental protocols were approved by local Institutional Biomedical Review Boards (ethics numbers: *14–0741, 11–0228, and 13–3613, 15–1626*) and performed in accordance with the relevant guidelines. Study characteristics of the cohorts are summarized in *Table 1*.

### PBMC isolation

PBMC were isolated from apheresis product by centrifugation ($\times$1200 g for 15 min at room temperature) on a Ficoll-Paque density gradient (GE Healthcare Life Sciences Ficoll-Paque Plus). Briefly, the apheresis product was diluted 1:2.5 in 2% FBS/PBS and 30 ml diluted product was overlaid onto 15 ml of Ficoll-Plaque in a 50 ml Falcon tube prior to centrifugation. PBMCs were harvested, then washed three times in 2% FBS/PBS. PBMC were counted and then frozen (*Goonetilleke et al., 2006*).

## Quantitative viral outgrowth assay (QVOA) and infectious units per million (IUPM)

QVOAs were performed as previously described (*Crooks et al., 2015*). Briefly, ~5 × 10$^7$ million resting CD4+ T cells, defined as CD4$^+$ CD45$^+$ CD3$^+$ CD69$^-$ CD25$^-$ CD8$^-$ CD14$^-$ HLA-DR$^-$, were isolated via negative selection from PBMC (*Keedy et al., 2009*). CD4+ T cells were maximally stimulated in limiting dilutions with phytohemagglutinin (PHA) (Remel-PHA; Thermo Scientific), IL-2, and irradiated CD8-depleted PBMC from a seronegative donor, which were added twice to cultures to support virus propagation. The CD8-depleted PBMC were obtained from seronegative donors previously screened to ensure adequate CCR5 expression. Culture supernatants were collected on days 15 and 19 and assayed for p24 expression by ELISA (ABL, Rockville, MD). Only cultures that contained an equivalent or greater level of p24 antigen on day 19 compared with day 15 were scored as positive. A maximum likelihood method was used on day 15 data to estimate the frequency of latent HIV infection, reported as bias-corrected Infectious Units Per Million (IUPM) (*Supplementary file 1*; *Trumble et al., 2017*). For 10 participants, longitudinal IUPM measurements were available. IUPM values measured in the 2 years prior to initial T cell mapping were averaged (range of IUPM values: 1–6, mean number of IUPM values/participant: 2). The limit of detection was defined as ≤1 p24 positive well (range 12–36 cultures, mean 18 cultures) at the highest input of cells (2.5 million cells).

## Replication-competent reservoir HIV-1 virus sequencing

Supernatants from end point dilutions of p24 positive wells from the QVOA were sequenced for replication-competent virus. Twenty-two participants had supernatants sequenced by nearly full-length sequencing, and one participant had supernatant sequenced by Primer ID. For seven participants, longitudinal p24 supernatants were available (2–4 time points over 4.4 years). In two participants, no sequencing was possible as no p24 supernatants were identified in the QVOA; in these assays IUPM was calculated as below the threshold of detection.

## Primer ID sequencing for target regions

A multiplexed Primer ID library that covered partial *gag*, integrase coding domain, V1/V2 region, partial gp41 and partial *nef* region was constructed using the previous protocol (*Zhou et al., 2015*; *Dennis et al., 2018*). In brief, viral RNA was extracted from culture supernatants using QIAmp viral mini kit (Qiagen). cDNA was synthesized using a cDNA primer mixture targeting the above regions with a block of random nucleotides in each cDNA primer serving as the Primer ID, and SuperScript III reverse transcriptase (ThermoFisher). After two rounds of purification of cDNA, we amplified the cDNA and attached Illumina adaptor sequences to the PCR products. Gel-purified libraries were pooled and sequenced using MiSeq 300 bp pair-end sequencing (Illumina). The TCS pipeline (https://github.com/SwanstromLab/PID) was used for de-mutiplexing of the sequenced regions and construct Primer ID Template Consensus Sequences (TCSs) for each sequenced viral template.

## Nearly full-length sequencing

Viral RNA was converted to cDNA using Superscript III reverse transcriptase and an oligo(dT) primer. The 3' and 5' half genomes were amplified in separate PCR reactions using barcoded primers, and the PCR products were pooled, and gel purified. The SMARTbell Template Prep Kit (PacBio) was used to add adaptors to amplicons, and amplicons were then submitted for PacBio sequencing (movie time of 10 hr). Sequences were demultiplexed by barcode, and then analyzed using the PacBio Long Amplicon Analysis (LAA) package. Sequences were aligned separately using MUSCLE v3.8.31 and in-house scripts were used to examine the frequencies of mutations (*Edgar, 2004a*; *Edgar, 2004b*). Neighbor-joining phylogenetic trees were built using Ninja v1.2.2 (http://wheelerlab.org/software/ninja/), and approximate maximum likelihood trees were built using FastTree 2.1 (*Price et al., 2010*).

After aligning using MUSCLE and trimming of positions with indels, we performed two-by-two comparisons of the sequences at 5HG and 3HG and calculated the percent of comparisons with 0 to 3 nucleotide differences among the total comparisons. We chose one nucleotide difference as the clonality cut-off as we estimated that the QVOA and the sequencing assay might generate two substitutions per HIV genome. Using this cut-off, the percent of clonality was defined as the percentage

of two-by-two comparisons yielding 0 to 1 nucleotide differences among all the comparisons. The Ruby module viral_seq was used to calculate the pair-wise diversity of sequences (https://rubygems.org/gems/viral_seq).

PacBio sequencing proficiency was confirmed by the strong correlation between participant IUPM and the number of full-length sequences obtained, both of which are determined by the number of p24+ ELISA wells in the QVOA assay (r = 0.755, p<0.0001), Spearman Rank, 24 pairs, (*Figure 3—figure supplement 2*).

## Droplet digital polymerase chain reaction (ddPCR)

The concentration of HIV-1 DNA copies in each sample was determined using the following primers and probes spanning HXB2 coordinates 684–810 (*Malnati et al., 2008*): LTRgagF: 5'-TCTCGACG-CAGGACTCG-3', LTRgagR: 5'- TACTGACGCTCTCGCACC- 3', and LTRgagProbe: 5'/56-FAM/CTC TCTCCT/ZEN/TCTAGCCTC/31ABkFQ/- 3'. Cell concentration was determined using the following RPP30 primers and probes: RPP30-F: 5'- GATTTGGACCTGCGAGCG-3', RPP30-R: 5'-GCGGCTGTC TCCACAAGT-3', RPP30-Probe: 5'-/56-FAM/CTGACCTGA/ZEN/AGGCTCT/31ABkFQ/- 3' (*Hindson et al., 2011*). LTRgag ddPCR reactions contained the following: 10 µl ddPCR Supermix for Probes (no dUTP) (Bio-Rad), 1 µl each primer (LTRgagF and LTRgagR) at a concentration of 20 µM, 0.35 µl of LTRgag probe at a concentration of 20 µM, 6.65 µl DEPC-treated water (Thermo Fisher Scientific), and 3 µl of undiluted DNA for a total volume of 22 µl. RPP30 ddPCR reactions contained the following: 10 µl ddPCR Supermix for Probes (no dUTP) (Bio-Rad), 1 µl each primer (RPP30-F and RPP30-R) at a concentration of 20 µM, 0.35 µl of RPP30 probe at a concentration of 20 µM, 8.65 µl DEPC-treated water (Thermo Fisher Scientific), and 1 µl of DNA diluted 1:100 in DEPC-treated water. 8E5 DNA was used as a positive control and no-template controls were included in each run. Samples were run in duplicate and the values averaged.

Droplets were generated using the QX200 Automated Droplet Generator (Bio-Rad), in a final volume of 40 µl. Following droplet generation, plates were sealed with a pierceable foil seal and immediately subjected to the following PCR conditions on a C1000 Touch Thermal Cycler with 96-Deep Well Reaction Module (Bio-Rad): 95°C for 10 min, then 45 cycles of 95°C for 30 s followed by 60°C for 1 min, a final extension of 98°C for 10 min and a 4°C hold. Following thermal cycling, plates were read on the QX200 Droplet Reader (Bio-Rad) with the following set-up: Experiment = Rare Event Detection, Mix = ddPCR Supermix for Probes (no dUTP), Target 1 = FAM, Target 2 = VIC.

Analysis of ddPCR data used QuantaSoft (Bio-Rad) with thresholds for the detection of each target set using the negative controls. The same threshold was used for all reactions of a particular target (either LTRgag or RPP30) in a given run. The concentration of HIV-1 copies / µl of sample was calculated as follows: Quantasoft LTRgag concentration ×(total volume of PCR reaction/volume of DNA added). The concentration of cells/µl of sample was calculated as follows: Quantasoft RPP30 concentration x (total volume of PCR reaction/volume of DNA added) x dilution factor x 0.5.

## DNA extraction

Total DNA was extracted from purified resting CD4+ T cells using the DNeasy Blood and Tissue Kit (Qiagen), following the manufacturer's instructions with the following modifications. DNA was eluted from the column using 200 µl of DEPC-treated water (Thermo Fisher Scientific). Following a one-minute centrifugation at 6000 x *g*, the eluate was re-applied to the membrane, incubated for 1 min at room temperature, and re-eluted. DNA was stored at −20°C until use.

## Amplification of *gag* and *nef*

A nested PCR approach was used to generate partial *gag* and *nef* amplicons. For amplification of partial *gag* amplicons, the first-round PCR primers were Gag_F_Outer_1: '5- GGACGGCGACTGG TGAGTACG −3' (HXB2 coordinates 732–752) and Gag_R_Outer: 5' – CCAATTCCCCCTATCA TTTTTGGTTTCC −3' (HXB2 coordinates 2404–2377). Each reaction contained 14.8 µl of DEPC-treated water, 2 µl of 10 × High Fidelity Buffer, 0.8 µl of 50 µM MgSO$_4$, 0.4 µl of 10 µM dNTP mix, 0.4 µl of forward and reverse primer at 20 µM and 5U of Platinum *Taq* DNA Polymerase High Fidelity (Thermo Fisher Scientific). One copy of HIV-1 DNA was added to each reaction contained, based on ddPCR estimates. A no template control was included in each run. First round thermal cycling conditions were as follows: an initial denaturation of 2 min at 98°C, followed by 35 cycles of: 15 s at 98°C

for denaturation, 30 s at 56°C for annealing, and 2 min 30 s at 72°C for extension, followed by a final extension at 72°C for 10 min and a 4°C hold. Second round PCR primers were as follows: Gag_F_Inner_2: 5′ – CTTTTGACTAGCGGAGGCTAGAAGG −3′ (HXB2 coordinates: 759–783) and Gag_R_Inner_3: 5′- GAGCTTCCCTTAGCTGACCCTCTAC −3′ (HXB2 coordinates: 2319–2295). Second round reactions were set up as described above, and 1 µl of first-round PCR product. Second round thermal cycling conditions were same as those described above, however, with an annealing temperature of 55°C.

A nested PCR approach was also used to generate *nef* amplicons. First round PCR primers were: U5-623F: 5′ -AAATCTCTAGCAGTGGCGCCCGAACAG −3′ (HXB2 coordinates 623–634) and U5-601R: 5′-TGAGGGATCTCTAGTTACCAGAGTC-3′ (HXB2 coordinates 9686–9662). Reaction components were identical to those described above. One copy of HIV-1 DNA was added to each reaction and a no-template control was included in each run. First round thermal cycling conditions were as follows: an initial denaturation of 2 min at 92°C, followed by 10 cycles of 10 s at 92°C for denaturation, 30 s at 60°C for annealing, 10 min at 68°C for extension, then 20 cycles of 10 s at 92°C for denaturation, 30 s at 55°C for annealing, 10 min at 68°C for extension, followed by a final extension of 10 min at 68°C, and a 4°C hold. The second-round primers were as follows: 4653F: 5′-CCCTACAATCCCCAAAGTCAAGGAG-3′ and OFM19: 5′- GCACTCAAGGCAAGCTTTATTGAGGCTTA-3′. Each reaction consisted of 34.8 µl of DEPC-treated water (Thermo Fisher Scientific), 5 µl of 10 × High Fidelity Buffer, 2 µl of 50 µM MgSO$_4$, 5 µl of 0.2 µM forward and reverse primers and 5 U of Platinum *Taq* DNA Polymerase High Fidelity (Thermo Fisher Scientific), and 2 µl of first round PCR product. Second round thermal cycling conditions were as follows: 2 min at 94°C for the initial denaturation, then 10 cycles of 15 s at 94°C for denaturation, 30 s at 55°C for annealing, and 8 min at 68°C for extension, then 25 cycles of 15 s at 94°C for denaturation, 30 s at 55°C for annealing, and 8 min at 68°C for extension with an additional 20 s of extension time added each cycle; a final 7 min at 68°C for extension, followed by a hold at 4°C. Second round PCR products were screened for positives on a 0.8% agarose gel stained with SYBR Safe DNA Gel Stain (Thermo Fisher Scientific), and visualized with a UV gel imager. Amplicons were sequenced using Sanger sequencing.

## HLA-associated polymorphisms methods

HLA-associated polymorphisms have previously been identified across the HIV-1 subtype B proteome via statistical association (*Carlson et al., 2012*). Briefly, HLA-*adapted* forms denote viral residues that are enriched among individuals expressing the restricting HLA (i.e. the inferred escape form), whereas HLA-*nonadapted* forms denote viral residues that are underrepresented among individuals expressing the restricting HLA (i.e. the inferred susceptible form) (*Carlson et al., 2012*). For each HIV-1 reservoir sequence, we identified all HIV codons under selection by one or more host HLA alleles (as defined in *Carlson et al., 2012*) and classified the autologous HIV residue as 'nonadapted', 'adapted' or 'possibly adapted' (where the latter extended to any variant other than the 'nonadapted' form). For each participant, a minimum bound of HLA-associated adaptation in the reservoir was estimated by computing the total proportion of HLA-associated sites exhibiting the 'adapted' form, while a maximum escape bound was estimated by including both 'adapted' and 'possibly adapted' forms. HLA-associated polymorphisms were analyzed at the HLA subtype (4-digit) level, where ambiguous allele calls were assigned to the most common allele in the list. Note that, for this analysis, Pol residue numbering begins at position −1 of protease.

## Peptides

18-mer peptides overlapping by 10 amino acids were synthesized (Sigma-Genosys, US) to match the clade B HIV-1 consensus sequence (https://www.hiv.lanl.gov/). The corresponding 18-mer autologous outgrowth reservoir virus peptide/s for all previously defined reactive 18-mers from the initial mapping ELISpot were also synthesized. In addition, optimal CD8+ T cell peptides predicted through LANL's epitope location finder ('ELF') software based on HLA-prediction software (https://www.hiv.lanl.gov/content/sequence/ELF/epitope_analyzer.html), and 18-mer peptides spanning stripes (see definition below) were synthesized. To cover variation in autologous OGV, a total of 656 peptides were synthesized.

## T cell mapping against HIV-1 clade B consensus sequence

Clade B consensus peptides (n = 386) were arranged into pools in a matrix format using the Peptide Portal program (SCHARP), adapting code from the Deconvolute this! Program (*Precopio et al., 2008*). Each peptide was repeated 3 × in the matrix peptide plate. Cryopreserved PBMCs were thawed and rested overnight before being placed in the ELISpot plates (Merck, Millipore) at $4 \times 10^5$ cells per well. Peptides were added to the plate with a final concentration of 4 µg/ml and incubated for 18–20 hr at 37°C, 5% $CO_2$. Coating, development (MabTech), and reading of ELISpot plates (AID Reader) has been described previously (*Goonetilleke et al., 2006*). Data from positive pools were then deconvoluted (*Roederer and Koup, 2003*) to identify candidate epitope peptides. Putative positive 18-mer peptide–specific T cell responses were confirmed in triplicate in a follow-up IFN-γ ELISpot with either 2- or $4 \times 10^5$ cells per well and a peptide concentration of 10 µg/ml, with 4 to 6 negative control wells (media only) and at least one positive control well (10 µg/ml PHA; Sigma-Aldrich) together. Positive T cell responses were defined as ≥12.5 SFU per million, ≥4 × background and no zero values in any replicate of antigen-stimulated wells.

## Evaluating T cell escape in epitopes previously defined as reactive in clade B consensus mapping

Testing OGV variants for T cell escape (replicates, controls, positivity criteria) was the same as for the initial mapping studies with the exception that peptides were tested at 10 µM. Escape variants were defined as eliciting a 50% decrease in the mean magnitude compared to the peptide that elicited the highest magnitude T cell response in IFN-γ ELISpot. Optimally defined CD8+ epitopes (8–11 aa) were defined as equal to, or greater than the T cell magnitude elicited by the 18-mer peptide (*Carlson et al., 2012*).

## Examining virus stripes and processing mutations

We observed clustered mutations in OGV sequence data outside of mapped T cell epitopes. We hypothesized that these regions may reflect pre-ART escape against T cell responses not detected using our matrix mapping approach. Regions where > 40% of the sequences contain non-synonymous mutations in the same position in the proteome were defined as 'stripes'. Twenty-four stripes were identified across participants. 18-mer peptides spanning stripes were synthesized and tested individually in ELISpot. As forecast, some T cell responses were detected. In total T cells recognized epitopes at 11/24 stripes, resulting in a total of 166 reactive T cell epitopes in 25 participants. Prediction software and HLA typing was used to screen virus sequences for the presence of antigen processing mutations (https://www.hiv.lanl.gov/content/immunology/variants/variant_search.html?db=ctl). No antigen processing mutations associated with reactive T cell epitopes were detected in the cohort.

## Entropy calculations

A Shannon entropy score (*Shannon, 1948*) was calculated for all reactive epitopes in this study, 18-mer and all experimentally confirmed optimal epitopes (*Supplementary file 4*). For each reactive epitope, the LANL sequence database was used to identify submitted Clade B sequences. Between 701 and 1936 sequences were analyzed per epitope. The Shannon Entropy-Two tool from the LANL database was used to identify the variation in protein sequence alignments (reactive epitope sequence vs. all known sequences), and an entropy score was calculated by averaging the entropy score for each amino acid position (https://www.hiv.lanl.gov/content/sequence/ENTROPY/entropy.html).

## Statistics

All statistical analyses were performed using Graph Pad Prism Version 8. Data were assumed to be non-normally distributed and as appropriate, two-tailed tests applied. A p value of $\leq 0.01$ was considered significant.

## Acknowledgements

We thank all the volunteers who participated in this study and all colleagues in respective labs who facilitated this study, including Catalina Ramirez, Erin Stuelke, Katherine Sholtis James, Brigitte Allard, Caroline Baker, John Schmitz and team, and the UCSF clinical team. We thank Genevieve Clutton for critical review of the manuscript. This study was funded by the University of North Carolina (UNC) Center for AIDS Research (P30 AI50410), the UNC Women's Interagency HIV Study (WIHS) (U01 AI103390), the University of California San Francisco WIHS (U01 AI034989), the National Institute of Allergy and Infectious Disease (U01 AI131310, R01 AI140970), with major contribution from CARE, a Martin Delaney Collaboratory of the National Institute of Allergy and Infectious Diseases, National Institute of Neurological Disorders And Stroke, National Institute On Drug Abuse and the National Institute Of Mental Health of the National Institutes of Health, (1UM1AI126619). Support was also provided by the BELIEVE Martin Delaney Collaboratory (UM1AI126617).

## Additional information

### Competing interests

Ronald Swanstrom: UNC is pursuing IP protection for Primer ID, and Ronald Swanstrom is listed as a coinventor and has received nominal royalties. The other authors declare that no competing interests exist.

### Funding

| Funder | Grant reference number | Author |
| --- | --- | --- |
| National Institute of Allergy and Infectious Diseases | P30 AI50410 | Ronald Swanstrom |
| National Institute of Allergy and Infectious Diseases | U01 AI103390 | Adaora A Adimora |
| National Institute of Allergy and Infectious Diseases | U01 AI034989 | Nadia Roan |
| National Institute of Allergy and Infectious Diseases | U01 AI131310 | Joanna A Warren<br>Shuntai Zhou<br>Yinyan Xu<br>Nilu Goonetilleke |
| National Institute of Allergy and Infectious Diseases | R01 AI140970 | Ronald Swanstrom |
| National Institute of Allergy and Infectious Diseases | UM1AI126617 | Daniel R MacMillan<br>Zabrina L Brumme |
| National Institute of Allergy and Infectious Diseases | 1UM1AI126619 | Joanna A Warren<br>Shuntai Zhou<br>Julia M Sung<br>JoAnn D Kuruc<br>Cynthia L Gay<br>David M Margolis<br>Nancie Archin<br>Nilu Goonetilleke |
| Canadian Institutes of Health Research | PJT-148612 | Daniel R MacMillan<br>Zabrina L Brumme |
| Canadian Institutes of Health Research | PJT-159625 | Daniel R MacMillan<br>Zabrina L Brumme |

The funders had no role in study design, data collection and interpretation, or the decision to submit the work for publication.

### Author contributions

Joanna A Warren, Formal analysis, Investigation, Methodology, Writing - original draft, Writing - review and editing; Shuntai Zhou, Formal analysis, Investigation, Methodology, Writing - original

draft; Yinyan Xu, Matthew J Moeser, Investigation; Daniel R MacMillan, Formal analysis; Olivia Council, Sarah Joseph, Data curation; Jennifer Kirchherr, Data curation, Investigation; Julia M Sung, Resources, Investigation, Writing - review and editing; Nadia R Roan, Resources, Writing - review and editing; Adaora A Adimora, Cynthia L Gay, Resources, Data curation; JoAnn D Kuruc, Resources, Data curation, Funding acquisition; David M Margolis, Resources, Formal analysis, Supervision, Funding acquisition, Methodology, Writing - review and editing; Nancie Archin, Resources, Data curation, Supervision, Writing - review and editing; Zabrina L Brumme, Conceptualization, Resources, Formal analysis, Supervision, Funding acquisition, Methodology, Writing - original draft, Writing - review and editing; Ronald Swanstrom, Data curation, Supervision, Writing - review and editing; Nilu Goonetilleke, Conceptualization, Resources, Data curation, Supervision, Funding acquisition, Writing - original draft, Writing - review and editing

### Author ORCIDs
Joanna A Warren http://orcid.org/0000-0003-0595-0390
Matthew J Moeser http://orcid.org/0000-0002-1368-5171
Nadia R Roan https://orcid.org/0000-0002-5464-1976
Zabrina L Brumme http://orcid.org/0000-0002-8157-1037
Nilu Goonetilleke https://orcid.org/0000-0003-2278-1656

### Ethics

Human subjects: Participants were enrolled with the following IRB approved studies: (1) CID 0819 - Apheresis Procedures to Obtain Leukocytes for Research Studies from HIV Positive Participants (08-1575), (2) The UNC Women's Interagency HIV Study (WIHS) (12-1660). Review and implementation of all protocols utilized for the collection of samples for this this analysis were approved by the University of North Carolina at Chapel Hill Biomedical Institutional Review Board (IRB) and the University of California at San Francisco IRB. All participants provided written informed consent. All experimental protocols were approved by local Institutional Biomedical Review Boards (ethics numbers: 14-0741, 11-0228, and 13-3613, 15-1626) and performed in accordance with the relevant guidelines.

### Decision letter and Author response
Decision letter https://doi.org/10.7554/eLife.57246.sa1
Author response https://doi.org/10.7554/eLife.57246.sa2

---

## Additional files

### Supplementary files
• Supplementary file 1. Size of the replication-competent reservoir in PLWH on ART (n = 25 participants) as measured by infectious units per million (IUPM) up to 2 years prior to T cell mapping date.

• Supplementary file 2. The number of near full-length sequences obtained from p24 antigen positive wells generated in quantitative viral outgrowth assays for each participant, and the pairwise diversity of sequences obtained (5' and 3' halves).

• Supplementary file 3. Percentage of clonal sequences obtained from p24 antigen positive wells generated in quantitative viral outgrowth assays for each participant (5' and 3' halves).

• Supplementary file 4. Summary of HLA-associated polymorphism by HIV-1 protein.

• Supplementary file 5. Summary of reactive T cell epitopes in study participants.

• Supplementary file 6. Raw data from studies evaluating viral T cell escape variants in the replication-competent reservoir.

• Supplementary file 7. HIV-1-specific T cell measurements and HIV reservoir size.

• Supplementary file 8. HIV-1-specific T cell responses do not correlated with the size of the HIV reservoir. Correlation between T cell breadth and summed magnitude of T cell response to HIV-1 protein in PLWH on ART (n = 25 participants, n = 166 epitopes), and adjusted for escape variants (n = 23 participants, n = 102 epitopes excluding 49 epitopes at which escape was observed),

measured by IFN-γ ELISpot and the size of the replication-competent reservoir as measured by infectious units per million (IUPM) using Spearman Rank.

• Supplementary file 9. Reactive T cell epitopes in PLWH on ART (n = 23 participants, 151 *total* mapped epitopes including 49 *escape* epitopes) are targeted by conserved immunogen vaccines. Escape in the HIV-1 reservoir is consistently lower in T cell epitopes that fall within conserved immunogen vaccines than mapped epitopes that fall outside of the immunogens.

• Supplementary file 10. Comparison of approaches to assess pre-ART escape in the latent HIV-1 reservoir.

• Transparent reporting form

## Data availability

Sequencing data have been deposited in Gen Bank under PRJNA666896, MT307344-MT308415 and MW054719-MW054856 All data generated (raw) or analyzed during this study are included in the manuscript and supporting files (supplementary files).

The following dataset was generated:

| Author(s) | Year | Dataset title | Dataset URL | Database and Identifier |
|---|---|---|---|---|
| Warren JA, Zhou S, Xu Y, Moeser M, MacMillan DR, Council O, Kirchherr J, Sung JM, Roan N, Adimora AA, Joseph S, Kuruc JD, Gay CL, Margolis DM, Archin NM, Brumme ZL, Swanstrom R, Goonetilleke N | 2020 | Primer ID sequencing for QVOA HIV sequence | https://www.ncbi.nlm.nih.gov/bioproject/666896 | NCBI BioProject, PRJNA666896 |

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
