## [Decision Letter]

**Acceptance summary:**

This work provides a detailed study to investigate the question of whether or not HIV harboured in the latent reservoir of patients well-controlled on HIV therapy can be recognized by their circulating T cells. In a carefully-performed study of 25 subjects, the authors use viral outgrowth culture and sequence analysis, followed by functional T cell assays, to show that most of the known epitopes remain subject to T cell recognition. This is an important result for HIV "cure" strategies, for which T cell recognition is likely to be an important component.

**Decision letter after peer review:**

Thank you for submitting your article "The HIV-1 latent reservoir is largely sensitive to circulating T cells" for consideration by *eLife*. Your article has been reviewed by three peer reviewers, and the evaluation has been overseen by a Reviewing Editor and Tadatsugu Taniguchi as the Senior Editor. The following individuals involved in review of your submission have agreed to reveal their identity: Steven Deeks (Reviewer #1); Sarah Rowland-Jones (Reviewer #3).

The reviewers have discussed the reviews with one another and the Reviewing Editor has drafted this decision to help you prepare a revised submission.

This manuscript describes a detailed study to investigate the question of whether or not HIV harboured in the latent reservoir of patients well-controlled on ART can be recognised by their circulating T cells. In a carefully-performed study of 25 subjects, the authors use viral outgrowth culture and sequence analysis, followed by functional T cell assays, to show that over ⅔ of known B clade epitopes remain subject to T cell recognition. This is an important result for HIV "cure" strategies, for which T cell recognition is likely to be an important component.

Summary:

Warren et al. report that the majority of detectable CD8+ T cell responses are able to recognize the latent viral reservoir in people living with HIV on antiretroviral therapy. The authors employ sequencing of outgrowth virus and epitope mapping of autologous CD8+ T cell responses across both consensus and observed variant sequences spanning the HIV proteome to assess mutational escape. On a per-person level, six of 23 persons had no observed escape mutations and two had complete escape, with the remaining 15 having escape from some but not all T cell responses, as defined by sequencing of outgrowth viruses and IFN-γ ELISpot assays. The work provides additional information for CD8+ T cell-based strategies to reduce or eliminate replication-competent proviral reservoirs, and is consistent with reports of T cell dysfunction as an additional mechanism of inadequate viral control and reports of CD8+ T cell contributions to limited viral control during and after cessation of ART.

1) It may be possible for virus mutations to arise during viral outgrowth, including reversion of mutations that incur fitness costs, which could have significant impacts on the study conclusions. Methods exist to determine full length proviral sequencing from the latent reservoir. OGV can be compared to matched proviral DNA sequencing from ex vivo resting CD4 T cells to address this. This seems critical to interpretation of the data.

2) Outgrowth in the absence of CD8 T cell pressure, as happens in the outgrowth assay, may preferentially disfavor escaped variants with fitness costs, which limits the ability for OGV sequencing alone to provide an accurate measurement of escape. Again, this speaks to the need for full length sequencing from the reservoir.

3) It is unclear why confirmation of CD8+ T cell responses using optimal epitopes was only achieved for a minority of the detected responses, which raises concerns about the majority of unconfirmed responses and whether they are suitable for inclusion in the analysis.

4) The use of ELISpot assays does not take into account flanking region mutations that are well known to alter antigen processing and serve as a mechanism of escape. Although this is perhaps beyond the scope of the study, it should be addressed as a caveat.

5) The authors acknowledge that their method for testing whether a variant epitope peptide escapes recognition or not is not completely comprehensive, and would miss – for example – variants that affect antigen processing or binding affinity (particularly the off-rate, as shown in reference 20 for the immunodominant B27 p24 epitope). Could they estimate – perhaps using data from the Molecular Immunology database? – the proportion of escape variants that might be missed using their methods?

6) The authors study four patients who initiated ART during acute HIV infection – the rest began therapy in chronic infection. One might predict that the length of the period of infection before initiating ART would influence the likelihood of developing escape mutations – do the authors have any data to support this hypothesis?

Revisions expected in follow-up work:

One reviewer felt that sequence data was needed to support this story's outlined in points 1 and 2, but other reviewers felt that this could be fairly limited and focused on the sequencing of the QVOA "non-escape" subjects. This more reasonable sequencing effort would help address the concerns of the one reviewer without requiring large amounts of sequence data.

---

## [Author Response]

Revisions for this paper:1) It may be possible for virus mutations to arise during viral outgrowth, including reversion of mutations that incur fitness costs, which could have significant impacts on the study conclusions. Methods exist to determine full length proviral sequencing from the latent reservoir. OGV can be compared to matched proviral DNA sequencing from ex vivo resting CD4 T cells to address this. This seems critical to interpretation of the data.One reviewer felt that sequence data was needed to support this story's outlined in points 1 and 2, but other reviewers felt that this could be fairly limited and focused on the sequencing of the QVOA "non-escape" subjects. This more reasonable sequencing effort would help address the concerns of the one reviewer without requiring large amounts of sequence data.

We believe that both our assay design and published data are incompatible with in vitro reversion. In our study, sequencing was performed on end-point dilution of virus i.e. on average, a single virus-infected cell. A numbered detailed response is provided below. In summary, while we have performed additional proviral DNA sequencing to address the reviewer’s concerns (detailed in #3) we do not consider it critical for interpretation of these data.

1) We are sequencing outgrowth virus from, on average, a single infected cell seeded in 2.5 million resting CD4 T cells. The culture period is 15 days. The effective population size in each well of the QVOA is n=1. Such a low effective population size means that the first mutation occurring in wells is stochastic or random. Deterministic models which assume a large effective population size do not apply here (reviewed in (*1*)). As one mutation occurs over a HIV cycle, in a 15-day culture we estimate that 4-5 cycles will occur. Therefore 5 mutations, initiated by a random mutation, are expected over the culture period.

Putting this information together, if reversion were to occur to a level to skew our study conclusions, then:

i) The first or second “stochastic” mutation would need to occur at the same or closely located positions in a ~10,000 bp HIV genome in not one, but multiple QVOA wells.

ii) Further, to impact our results this stochastic mutation would need to consistently dominate the viral population in no more than 4 virus cycles in multiple QVOA wells. iii) Further still, the mean T cell breadth was 7 in this study i.e. 7 T cell epitopes across the entire HIV proteome. Therefore, to skew our results five mutations generated over 15 days would need to induce non-synonymous reverting changes in multiple epitopes as well as dominate the viral population.

We would hope the reviewers would agree that the likelihood of i-iii occurring is extremely low to impossible.

This scenario, that is in vitro reversion arising for a single infectious molecular clone (IMC) was tested in Song et al. (Feng Gao, Duke). The authors infected CD4 T cells with an IMC (note, the effective population size would be >>> than endpoint dilution). Cells were cultured for approximately 15 days and virus sequenced using single genome amplification as used in this study. See Figure 8 from Song et al., 2012 for the highlighter plot. SGA sequencing (n=47) showed rare mutations (average = 1.3 nucleotide changes per genome, (range 0-4) in 3’ SGA half (4685bp) and no evidence of directional selection. This paper is now cited in the manuscript with text that discusses the low probability of in vitro reversion.

2) in vivo studies of reversion also suggest that in vitro reversion in QVOA is highly unlikely. In transmission studies including mother-to-child studies in which the recipient lacks the selecting HLA, documented reversion has required months to years (*2-6*). Reversion to consensus in acute infection studies lacking T cell epitope data has been described to occur more rapidly, though still over many weeks to months (*7, 8*). Please note that in contrast to QVOA, in acute infection the effective population size of HIV is very high bringing in deterministic models of virus mutation which include the effects of recombination where more rapid virus evolution would be predicted.

3) We hope that #1 and #2 above, make clear why we do not consider additional generation of proviral sequencing necessary for interpretation of data in this paper. However, we agree that inclusion of proviral sequencing is very interesting and further rounds out this paper. Note, our additional data are restricted to n=2 participants with sequencing of genomic regions, not full-length sequencing. This reflects the extraordinary cost of DNA sequencing approaches, cell limitations and genuine technical difficulty of these approaches. For example, generation of these data took 11 weeks and cost $15,000 in reagents.

In brief, we isolated DNA from 5 million resting memory CD4^+^ T cells (50million PBMC) and sequenced near-full *nef* and *gag* regions in 00728 and 00720. Longitudinal reservoir sequencing was available for both participants providing good numbers of comparator outgrowth viruses to address the reviewer’s concerns. The highlighter plot shown in Figure 4—figure supplement 2A shows clearly that the intact DNA sequences (no stop codon) show very high sequence identity with OGVs with no evidence of reversion at or around T cell epitopes (gray boxes). Please note that while identical gene-level sequences were observed, the parent outgrowth viruses were not clonal. As such, the detection of identical DNA and outgrowth virus gene sequences does not represent a dominant clonal sequence, that could skew data.

In summary, consistent with modeling estimates explained in #1 above, there was no evidence of reversion or other selection occurring in HIV DNA not detectable OGVs. These data are discussed in the manuscript, new data included in Figure 4—figure supplement 2 and the Materials and methods.

2) Outgrowth in the absence of CD8 T cell pressure, as happens in the outgrowth assay, may preferentially disfavor escaped variants with fitness costs, which limits the ability for OGV sequencing alone to provide an accurate measurement of escape. Again, this speaks to the need for full length sequencing from the reservoir.

Please see our response above. Given there is on average, a single virus-infected cell in a well, there are no a priori variants that are favored and stochastic, not deterministic, mutation governs the changes in vitro. A random mutation in cycle one that has a fitness benefit would need to sweep through the population in 4-5 virus cycles. Given the consistency of our sequencing data across wells at T cell epitopes, this favorable fitness mutation would also need to randomly occur and then overtake the virus population in 4 cycles in the majority of independent cultures in our assay system. This would need to occur at multiple T cell epitopes. We believe the probably of such events occurring is near impossible. In summary, the absence of CD8 T cell selection pressure has very little effect of the virus population undergoing stochastic changes in this short-term culture.

3) It is unclear why confirmation of CD8+ T cell responses using optimal epitopes was only achieved for a minority of the detected responses, which raises concerns about the majority of unconfirmed responses and whether they are suitable for inclusion in the analysis.

We are sorry to appear so contrary, but we must also strongly disagree with the reviewer’s comment here. A documented T cell response to an 18mer peptide is not “unconfirmed” and *all* T cell data presented are appropriate for analysis. Our response below is divided into sections A and B. “A” details why a documented T cell responses to a longer peptide is valid and “B” details why in this study did not subsequently map all optimal epitopes.

A) There are >20 years of literature from scores of laboratories to show that longer peptides can be used to successfully map T cell responses and to describe T cell escape. To our knowledge, there are no reports in the literature of a measured T cell response to a longer peptide being a false positive, not also detectable by a shorter peptide. The biology here is that the core epitope of the longer peptides can still sit in the HLA pocket and overhangs do not significantly interfere with T cell recognition. Studies using longer peptides span infectious disease through to cancer, T cell mapping in mouse (9), NHP (10) and humans (11). In HIV, the Walker-Harvard (11) , Goulder-Oxford (12, 13), McElrath-FHCRC and Brander-IrsiCaixa (14) have all routinely mapped HIV specific T cell responses using longer overlapping peptides; this includes clinical testing of T cell vaccines (15, 16). Indeed, the majority of T cell responses in HIV research are reported in terms of responses to overlapping longer (15-20mer) peptide pools spanning a HIV protein, often Gag (17). We and others have used longer peptide to document HIV escape in multiple studies (18-20). Last, we would note that a high proportion of published studies in HIV that examine virus escape, no longer empirically T cell responses and simply rely on prediction methods.

Our Materials and methods section clearly details the rigorous criteria applied for assessing a T cell response and mapping epitopes i) all peptides are tested independently in quadruplicate and only considered reactive when 4x greater than the mean of 6 mock wells) ii) all reactive peptides were then independently re-tested again in quadruplicate in all assays assessing variants. We went further and then generated the corresponding autologous peptides, again testing both the clade B peptide alongside another – this compares with vast majority of HIV studies that map using only consensus peptide sets. Across our dataset we did not observe any discordant data in independent testing. As we note in our paper, our study is the largest proteome-wide mapping study in PLWH on ART to date. We provide all T cell data in an attached spreadsheet to the study as a resource for the field.

B) HLA diversity prevents optimal epitope mapping of every new T cell response. To date, thousands of HLA I alleles have been identified and the binding motifs are only known for a small subset of HLA alleles. This is consistent with previous studies in which no optimal epitope was known/predicted in over 60% of mapped reactive HIV peptides (19-21). To empirically identify the optimal epitope and then define the HLA restriction firstly, 8- 9- and 10- peptides overlapping by 7-9 amino acids need to be synthesized for each reactive longer peptide. Once the shortest reactive epitope has been defined then restriction should then be confirmed using antigen presentation from a cell line expressing a single HLA. To determine optimal CD8+ T cell epitopes for all CD8+ T cell responses in our studies, we estimate that we would need to generate 3,000-4,000 autologous peptides ($60-80K in peptide costs alone). Testing would require additional large blood draws of all our study participants.

To generate the most informative and detailed dataset while maintaining reasonable costs, we used bioinformatics approaches combine with HLA typing to predict the optimal epitope for common HLA alleles in the human population. Of 166 epitopes, bioinformatic algorithms predicted >120 optimal epitopes. Empirical testing showed that only 20% of these predictions were correct, demonstrating that bioinformatics approaches are not perfect.

Summary: With respect, we feel the reviewer is holding us to an unrealistic and more important, an unnecessary standard. We reiterate that all T cell responses described in this paper were confirmed and comply with the most rigorous standards for measurement in our field.

4) The use of ELISpot assays does not take into account flanking region mutations that are well known to alter antigen processing and serve as a mechanism of escape. Although this is perhaps beyond the scope of the study, it should be addressed as a caveat.5) The authors acknowledge that their method for testing whether a variant epitope peptide escapes recognition or not is not completely comprehensive, and would miss – for example – variants that affect antigen processing or binding affinity (particularly the off-rate, as shown in reference 20 for the immunodominant B27 p24 epitope). Could they estimate – perhaps using data from the Molecular Immunology database? – the proportion of escape variants that might be missed using their methods?

We agree that our empirical method will miss flanking variants that affect antigen processing or binding affinity and is noted as a limitation in our Discussion. As per the reviewer’s suggestion we looked for flanking / processing mutations to reactive epitopes on the LANL database and found none detailed.

The bioinformatic approach developed by Dr Brumme, specifically “possible adaptations” extends to processing and binding affinity ns changes previously associated with specific HLA in untreated infection. Consistent with this, the predicted level of escape from “possibly adapted” was higher (48%) than our empirical testing identifying variants, including binding affinity variants, we may have missed. We have revised the Discussion to emphasize this point.

6) The authors study four patients who initiated ART during acute HIV infection – the rest began therapy in chronic infection. One might predict that the length of the period of infection before initiating ART would influence the likelihood of developing escape mutations – do the authors have any data to support this hypothesis?

We agree with the reviewer that length of untreated infection will influence escape prevalence, consistent with previous studies report that early treatment restricts the virus population diversity at an early stage and limits the emergence of new immune-escape variants (reviewed in doi: 10.1097/COH.0000000000000120).

In our cohort, no difference in the likelihood of developing escape mutations was observed between participants treated in acute (n=3 with sufficient OGV data) or chronic HIV-1 infection (Mann-Whitney two tailed test, p=0.675). This is noted in the manuscript, however we do not consider we have sufficient study numbers to further emphasize this point.

**References**

A. J. Brown, D. D. Richman, HIV-1: gambling on the evolution of drug resistance? *Nature medicine* 3, 268-271 (1997).

A. J. Leslie *et al.*, HIV evolution: CTL escape mutation and reversion after transmission. *Nature medicine* 10, 282-289 (2004).

S. J. Kent, C. S. Fernandez, C. J. Dale, M. P. Davenport, Reversion of immune escape HIV variants upon transmission: insights into effective viral immunity. *Trends in microbiology* 13, 243-246 (2005).

H. Crawford *et al.*, Compensatory mutation partially restores fitness and delays reversion of escape mutation within the immunodominant HLA-B*5703-restricted Gag epitope in chronic human immunodeficiency virus type 1 infection. *Journal of virology* 81, 8346-8351 (2007).

K. Gounder *et al.*, High frequency of transmitted HIV-1 Gag HLA class I-driven immune escape variants but minimal immune selection over the first year of clade C infection. *PloS one* 10, e0119886 (2015).

C. F. Thobakgale *et al.*, Impact of HLA in mother and child on disease progression of pediatric human immunodeficiency virus type 1 infection. *Journal of virology* 83, 10234-10244 (2009).

T. M. Allen *et al.*, Selective escape from CD8+ T-cell responses represents a major driving force of human immunodeficiency virus type 1 (HIV-1) sequence diversity and reveals constraints on HIV-1 evolution. *Journal of virology* 79, 13239-13249 (2005).

V. Novitsky *et al.*, Timing constraints of in vivo gag mutations during primary HIV-1 subtype C infection. *PloS one* 4, e7727 (2009).

B. Ondondo *et al.*, Novel Conserved-region T-cell Mosaic Vaccine With High Global HIV-1 Coverage Is Recognized by Protective Responses in Untreated Infection. *Molecular therapy : the journal of the American Society of Gene Therapy* 24, 832-842 (2016).

A. T. Jones *et al.*, HIV-1 vaccination by needle-free oral injection induces strong mucosal immunity and protects against SHIV challenge. *Nature communications* 10, 798 (2019).

M. M. Addo *et al.*, Comprehensive epitope analysis of human immunodeficiency virus type 1 (HIV-1)-specific T-cell responses directed against the entire expressed HIV-1 genome demonstrate broadly directed responses, but no correlation to viral load. *Journal of virology* 77, 2081-2092 (2003).

P. Kiepiela *et al.*, CD8+ T-cell responses to different HIV proteins have discordant associations with viral load. *Nature medicine* 13, 46-53 (2007).

P. Kiepiela *et al.*, Dominant influence of HLA-B in mediating the potential co-evolution of HIV and HLA. *Nature* 432, 769-775 (2004).

B. Mothe *et al.*, Definition of the viral targets of protective HIV-1-specific T cell responses. *Journal of translational medicine* 9, 208 (2011).

A. Guimaraes-Walker *et al.*, Lessons from IAVI-006, a phase I clinical trial to evaluate the safety and immunogenicity of the pTHr.HIVA DNA and MVA.HIVA vaccines in a prime-boost strategy to induce HIV-1 specific T-cell responses in healthy volunteers. *Vaccine* 26, 6671-6677 (2008).

N. Goonetilleke *et al.*, Induction of multifunctional human immunodeficiency virus type 1 (HIV-1)-specific T cells capable of proliferation in healthy subjects by using a prime-boost regimen of DNA- and modified vaccinia virus Ankara-vectored vaccines expressing HIV-1 Gag coupled to CD8+ T-cell epitopes. *Journal of virology* 80, 4717-4728 (2006).

A. Masemola *et al.*, Hierarchical targeting of subtype C human immunodeficiency virus type 1 proteins by CD8+ T cells: correlation with viral load. *Journal of virology* 78, 3233-3243 (2004).

V. V. Ganusov *et al.*, Fitness costs and diversity of the cytotoxic T lymphocyte (CTL) response determine the rate of CTL escape during acute and chronic phases of HIV infection. *Journal of virology* 85, 10518-10528 (2011).

N. Goonetilleke *et al.*, The first T cell response to transmitted/founder virus contributes to the control of acute viremia in HIV-1 infection. *The Journal of experimental medicine* 206, 1253-1272 (2009).

M. K. Liu *et al.*, Vertical T cell immunodominance and epitope entropy determine HIV-1 escape. *The Journal of clinical investigation* 123, 380-393 (2013).

M. Altfeld *et al.*, Enhanced detection of human immunodeficiency virus type 1-specific T-cell responses to highly variable regions by using peptides based on autologous virus sequences. *Journal of virology* 77, 7330-7340 (2003).